# PINE: Pruning Boosted Tree Ensembles with Conformal In-Distribution Prediction Equivalence

**Haruki Yajima** [1]   **Yusuke Matsui** [1]

## Abstract

Tree ensembles are machine learning models with strong predictive performance and interpretability, and remain widely used for tabular data. Standard pruning methods for tree ensembles typically optimize an accuracy-compression trade-off and may change a subset of predictions, potentially compromising decision consistency. Faithful pruning methods address this issue by preserving prediction equivalence over the entire input space, but this requirement leads to lower compression ratios. We propose **PINE**, a pruning method that provides strong guarantees within an in-distribution region. PINE preserves prediction equivalence within this region and controls the region size using a single parameter $\alpha$ via conformal calibration. Experiments on 12 public tabular datasets show that PINE improves the compression ratio by up to $30\%$ while preserving predictions at a comparable level to existing faithful pruning methods.

## 1. Introduction

Tree-based ensemble methods, such as Random Forests (Breiman, 2001), Gradient-Boosted Decision Trees (GBDT) (Friedman, 2001), and XGBoost (Chen & Guestrin, 2016), often achieve strong performance on tabular data (Grinsztajn et al., 2022). Their interpretability has also enabled adoption in high-stakes domains such as medicine and finance (Caruana et al., 2015; Bussmann et al., 2021). Increasing the number of trees is well known to improve accuracy (Biau & Scornet, 2016; Probst & Boulesteix, 2018; Buschjäger & Morik, 2021). However, larger ensembles increase inference time and memory footprint (Lucchese et al., 2017; Ye et al., 2018; Chen et al., 2022), and can make downstream verification of robustness and fairness more challenging (Kantchelian et al., 2016;

Devos et al., 2021; Ranzato et al., 2021; Calzavara et al., 2023). Consequently, post-hoc ensemble compression via ensemble pruning is an important problem.

When pruning a deployed tree ensemble, faithful pruning methods (Vidal & Schiffer, 2020; Emine et al., 2025) are a natural option because they preserve predictions before and after pruning for all inputs, but this strong requirement often leads to lower compression. These methods focus on fidelity, i.e., the fraction of inputs on which the pruned model's predictions match those of the original model, rather than test accuracy. Low fidelity means that many predictions differ before and after pruning, which can affect model-based decision making (Milani Fard et al., 2016; Bahri & Jiang, 2021). Such changes are especially costly in high-stakes settings, including downstream workflows triggered by model outputs (Srivastava et al., 2020), robustness or fairness checks built around the model (Devos et al., 2021; Calzavara et al., 2023), and explanations or recourse suggestions derived from its decisions (Rudin, 2019; Parmentier & Vidal, 2021). However, by enforcing prediction equivalence even for out-of-distribution (OOD) inputs that are rarely observed in practice, faithful pruning can yield lower compression than standard accuracy-oriented pruning methods.

We propose **Pruning with In-distribution Equivalence (PINE)**, a pruning method for decision tree ensembles that enables higher compression by guaranteeing prediction equivalence only within an in-distribution region. With a single hyperparameter $\alpha$, PINE defines an in-distribution region that includes future inputs sampled from the same distribution with probability at least $1 - \alpha$. PINE guarantees exact prediction equivalence before and after pruning within this region. This design makes it possible to systematically control the fidelity-compression trade-off via $\alpha$. In the toy example (Figure 1), PINE prunes 23% more trees than a faithful pruning baseline while preserving comparable fidelity. Empirically, across 12 public tabular datasets, PINE improves the compression ratio by up to $30\%$ over faithful pruning baselines while maintaining comparable fidelity, and it substantially improves fidelity over accuracy-oriented pruning methods at similar test accuracy.

---

[1]The University of Tokyo. Correspondence to: Haruki Yajima <yajima@hal.t.u-tokyo.ac.jp>.

*Proceedings of the 43rd International Conference on Machine Learning*, Seoul, South Korea. PMLR 306, 2026. Copyright 2026 by the author(s).

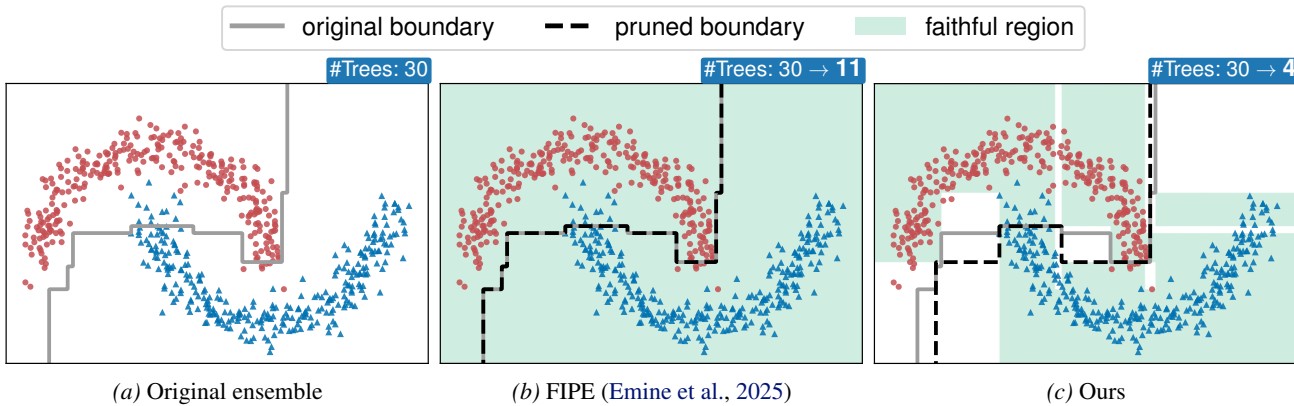

*(a)* Original ensemble  *(b)* FIPE (Emine et al., 2025)  *(c)* Ours

*Figure 1.* Comparison of pruning results between the baseline FIPE and the proposed PINE on a 2D synthetic dataset. The green faithful region indicates where prediction equivalence is guaranteed. (a) Decision boundary of a trained decision tree ensemble consisting of 30 trees. (b) FIPE (Emine et al., 2025) guarantees prediction equivalence with the original model over the entire input space. However, because it also preserves fine-grained boundaries in out-of-distribution regions such as the top-right corner, it can reduce the ensemble only from 30 trees to 11, resulting in lower compression. (c) PINE guarantees prediction equivalence only within a region aligned with the data distribution. By replacing fine-grained out-of-distribution boundaries with simpler ones, it reduces the ensemble from 30 trees to 4 while achieving comparable fidelity.

## 2. Related Work

**Accuracy-oriented pruning of tree ensembles.** Standard pruning methods for decision tree ensembles optimize an accuracy-compression trade-off (Margineantu & Dietterich, 1997; Zhang et al., 2006). Representative tree-level approaches select a subset of trees based on contribution and diversity (Lu et al., 2010; Li et al., 2012; Guo et al., 2018) or sparsify post-training tree weights via $L_1$ regularization (Buschjäger & Morik, 2023). Finer-grained compression methods include leaf refinement, which re-estimates leaf predictions (Ren et al., 2015), and methods that directly remove internal nodes or subtrees to increase the pruning rate (Liu & Mazumder, 2023; Devos et al., 2025). These approaches aim to reduce model size while tolerating some loss in accuracy, and they do not guarantee prediction equivalence with the original model.

**Faithful pruning of tree ensembles.** For decision tree ensembles, faithful pruning methods have been proposed to compress models while ensuring exact prediction equivalence before and after pruning. Born-Again Tree Ensembles (Vidal & Schiffer, 2020) exploit the fact that a tree ensemble partitions the input space into finitely many regions, and use dynamic programming to construct a single decision tree that is equivalent to the original ensemble. FIPE (Emine et al., 2025) identifies, via iterative optimization, the sparsest tree weights that preserve predictions of the original model. However, because these methods require prediction equivalence over the entire input space, their constraints also apply to out-of-distribution regions that are rarely observed, thereby limiting compression. We improve the fidelity-compression trade-off by restricting the guarantee region to an in-distribution region.

## 3. Preliminaries

### 3.1. Tree Ensembles

Here, we introduce decision tree ensembles, which aggregate predictions from multiple decision trees to produce a final prediction (Hastie et al., 2009). Let $\mathcal{X} \subseteq \mathbb{R}^p$ be a $p$-dimensional input space, and let $\mathcal{Y} = \{1, \ldots, C\}$ be the set of classes in a $C$-class classification problem. Consider a decision tree ensemble $\mathcal{T} = \{T_m\}_{m=1}^{M}$ consisting of $M$ decision trees. Each tree $T_m$ has finitely many leaves, with leaf index set $\mathcal{L}_m = \{1, 2, \ldots\}$. During training, each leaf $\ell \in \mathcal{L}_m$ is associated with a constant vector $\boldsymbol{v}_{m,\ell} \in \mathbb{R}^C$ representing per-class prediction scores. Let $\lambda_m : \mathcal{X} \to \mathcal{L}_m$ be the function that returns the leaf index reached by input $\boldsymbol{x} \in \mathcal{X}$ in tree $T_m$; then the prediction score of tree $T_m$ on $\boldsymbol{x}$ is $\boldsymbol{v}_{m,\lambda_m(\boldsymbol{x})}$. By construction of the decision tree, $\lambda_m(\boldsymbol{x})$ is uniquely determined for any input $\boldsymbol{x}$.

Under this structure, we define the prediction function of $\mathcal{T}$. Each tree $T_m$ has a non-negative weight $w_m^{(0)} \in \mathbb{R}_{\geq 0}$, and we denote the weight vector by $\boldsymbol{w}^{(0)} = (w_1^{(0)}, \ldots, w_M^{(0)})^\top$. The superscript $(0)$ indicates weights of the trained ensemble before pruning. The prediction score vector of the entire ensemble for input $\boldsymbol{x} \in \mathcal{X}$ is

$$\boldsymbol{F}(\boldsymbol{x}; \boldsymbol{w}^{(0)}) := \sum_{m=1}^{M} w_m^{(0)} \, \boldsymbol{v}_{m,\lambda_m(\boldsymbol{x})}, \qquad (1)$$

and the predicted class is defined as the one with the maximum score:

$$\hat{y}(\boldsymbol{x}; \boldsymbol{w}^{(0)}) := \operatorname*{argmax}_{c \in \mathcal{Y}} \big[ \boldsymbol{F}(\boldsymbol{x}; \boldsymbol{w}^{(0)}) \big]_c, \qquad (2)$$

where $\hat{y}(\boldsymbol{x}; \boldsymbol{w}^{(0)})$ denotes the ensemble prediction. In the case of ties in Equation (2), we choose the smallest class

index so that $\hat{y}$ is deterministic. Hereafter, we distinguish the original weights $\boldsymbol{w}^{(0)}$ from the post-pruning weights $\boldsymbol{w}$. Since Equation (1) is uniquely determined by $\boldsymbol{w}$, when the context is clear, we may refer to $\boldsymbol{w}$ as the model, meaning the weighted ensemble itself.

## 3.2. Faithful Pruning

FIPE (Emine et al., 2025) is a pruning method for decision tree ensembles that adjusts the ensemble weights $\boldsymbol{w}^{(0)}$ to find the sparsest weights $\boldsymbol{w}$ that preserve predictions for all inputs $\boldsymbol{x} \in \mathcal{X}$. This procedure is known as *faithful pruning* and is formulated as

$$\underset{\boldsymbol{w} \in \mathbb{R}_{\geq 0}^M}{\arg\min} \, \|\boldsymbol{w}\|_0 \tag{3a}$$

$$\text{s.t. } \hat{y}(\boldsymbol{x}; \boldsymbol{w}) = \hat{y}(\boldsymbol{x}; \boldsymbol{w}^{(0)}), \, \forall \boldsymbol{x} \in \mathcal{X}, \tag{3b}$$

where $\|\boldsymbol{w}\|_0$ denotes the number of non-zero weights. Trees with zero weight are removed, so minimizing $\|\boldsymbol{w}\|_0$ reduces the number of trees. Although $\mathcal{X}$ is continuous, tree ensembles partition $\mathcal{X}$ into finitely many cells on which $\hat{y}(\cdot)$ is constant; thus Equation (3) amounts to enforcing a large (but finite) set of equivalence constraints. As explicitly enumerating these constraints is intractable, FIPE solves the problem by iteratively applying `Pruner` and `Oracle`. Note that because argmax is scale-invariant, solutions are non-unique up to positive scaling; one may fix scale (e.g., $\sum_m w_m = 1$) without affecting pruning.

Specifically, `Oracle` searches for inputs whose predictions differ between the current weights $\boldsymbol{w}$ and the original weights $\boldsymbol{w}^{(0)}$, and `Pruner` updates weights to satisfy the constraints. Let $\boldsymbol{x}^\star$ be an input that changes the prediction, i.e., satisfies $\hat{y}(\boldsymbol{x}^\star; \boldsymbol{w}) \neq \hat{y}(\boldsymbol{x}^\star; \boldsymbol{w}^{(0)})$. We call such an input a *counterexample*. `Oracle` checks whether counterexamples exist for the current weights $\boldsymbol{w}$, and returns a set of counterexamples if they exist. This check can be formulated as selecting a combination of leaves that realizes a score change sufficient to alter the predicted class, and can be solved as a Mixed-Integer Linear Programming (MILP). FIPE adopts the MILP formulation of OCEAN (Parmentier & Vidal, 2021). If no counterexample is found, the current $\boldsymbol{w}$ is guaranteed to satisfy Equation (3b) over the entire input space. `Pruner` returns the sparsest weights $\boldsymbol{w}$ that satisfy prediction equivalence on the set of counterexamples found so far. FIPE solves Equation (3) by applying `Pruner` and `Oracle` until no new counterexamples are found.

## 3.3. Threshold Calibration via Split Conformal

We review split conformal prediction, which we use to control the size of the guarantee region in our method. Given a real-valued score function $s : \mathcal{X} \to \mathbb{R}$ on inputs $\boldsymbol{x} \in \mathcal{X}$, we consider the problem of choosing a threshold $\tau$ so that future inputs satisfy $s(\boldsymbol{x}) \leq \tau$ with a desired probability.

Split conformal prediction (also known as inductive conformal prediction) (Vovk et al., 2005; Shafer & Vovk, 2008; Lei et al., 2018; Angelopoulos & Bates, 2023) is a standard approach for this purpose.

Let $\mathcal{D}_{\text{cal}} = \{\boldsymbol{x}_i\}_{i=1}^n$ be a calibration set of inputs obtained i.i.d. from an unknown data distribution $P_0$. Fix a miscoverage level $\alpha \in (0, 1)$, i.e., the probability that a future input does not satisfy the threshold condition. In split conformal prediction, we calibrate the threshold $\tau(\alpha)$ as an order statistic corresponding to the $(1 - \alpha)$ quantile of the scores on $\mathcal{D}_{\text{cal}}$. Let $s_{(1)} \leq \cdots \leq s_{(n)}$ denote the order statistics of the calibration scores $\{s(\boldsymbol{x}_i)\}_{i=1}^n$. Here, the score function $s$ can be chosen by the practitioner. We set the threshold $\tau(\alpha)$ as

$$\tau(\alpha) := s_{(k)}, \, k := \lceil (n+1)(1 - \alpha) \rceil . \tag{4}$$

If $k = n + 1$, we interpret $\tau(\alpha) = +\infty$. Note that $\tau(\alpha)$ is non-increasing in $\alpha$, since increasing $\alpha$ corresponds to a lower $(1 - \alpha)$ quantile of the calibration scores.

Under the assumption that the calibration set $\mathcal{D}_{\text{cal}}$ and the future input are exchangeable (i.e., their joint distribution is invariant to permutations), split conformal provides a distribution-free coverage guarantee. Let $\boldsymbol{X}_{\text{new}}$ be a future input random variable taking values in $\mathcal{X}$. With a slight abuse of notation, we write $s(\boldsymbol{X}_{\text{new}})$ for the induced real-valued random variable. Then, the following inequality holds:

$$\mathbb{P}[s(\boldsymbol{X}_{\text{new}}) \leq \tau(\alpha)] \geq 1 - \alpha. \tag{5}$$

Therefore, at inference time, it suffices to check whether the observed input $\boldsymbol{x}_{\text{new}}$ satisfies $s(\boldsymbol{x}_{\text{new}}) \leq \tau(\alpha)$.

## 4. PINE: Pruning with In-Distribution Equivalence

We propose a new pruning method for decision tree ensembles, **Pruning with In-distribution Equivalence (PINE)**. As shown in Algorithm 1 and Figure 2, PINE builds on the FIPE (Emine et al., 2025) framework for faithful pruning, but provides a strong guarantee only for inputs within an in-distribution region $\mathcal{X}_{\text{ID}}(\alpha) \subseteq \mathcal{X}$: predictions before and after pruning are identical on $\mathcal{X}_{\text{ID}}(\alpha)$. PINE controls the size of $\mathcal{X}_{\text{ID}}(\alpha)$ using a single parameter $\alpha$, which enables systematic control of the fidelity-compression trade-off. Specifically, PINE defines the in-distribution region where prediction equivalence is guaranteed using a plausible score $s(\boldsymbol{x})$ and a threshold $\tau(\alpha)$ as

$$\mathcal{X}_{\text{ID}}(\alpha) := \{\boldsymbol{x} \in \mathcal{X} \mid s(\boldsymbol{x}) \leq \tau(\alpha)\}, \tag{6}$$

where smaller scores indicate more plausible inputs. We estimate the score function $s(\cdot)$ and train the decision tree ensemble using a dataset $\mathcal{D}_{\text{fit}}$. We calibrate the threshold

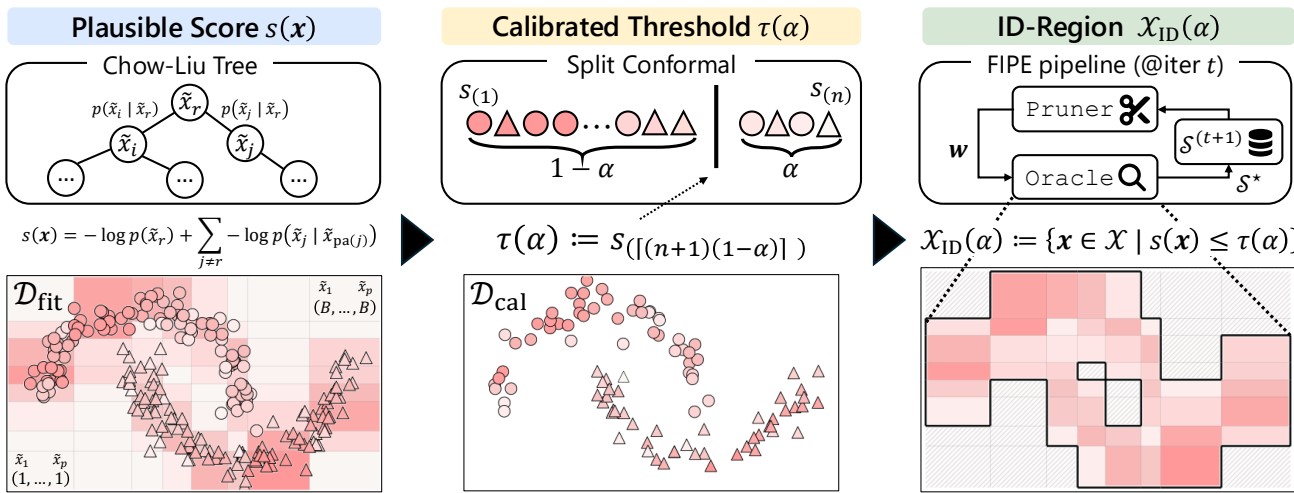

*Figure 2.* Overview of PINE. We calibrate an in-distribution region based on a plausible score $s(\cdot)$ via split conformal prediction, and search for counterexamples using an Oracle whose search is restricted to this region $\mathcal{X}_{\text{ID}}(\alpha)$. If no counterexample exists, exact prediction equivalence holds on $\mathcal{X}_{\text{ID}}(\alpha)$.

---

**Algorithm 1** Pruning with In-distribution Equivalence

---

**Require:** tree ensemble $\mathcal{T}$, original weights $\boldsymbol{w}^{(0)}$, miscoverage $\alpha$, fit set $\mathcal{D}_{\text{fit}}$, calibration set $\mathcal{D}_{\text{cal}}$
**Ensure:** pruned weights $\boldsymbol{w}$
1:  Fit plausible score function $s(\cdot)$ on $\mathcal{D}_{\text{fit}}$
2:  Calibrate threshold $\tau(\alpha)$ using $\mathcal{D}_{\text{cal}}$
3:  $\mathcal{S}^{(0)} \leftarrow \mathcal{D}_{\text{fit}}$
4:  $t \leftarrow 0$
5:  # At most $\mathcal{O}(e^{\tau(\alpha)})$ times (see Proposition 4.3)
6:  **repeat**
7:      # $\mathcal{O}(M + |\mathcal{S}^{(t)}|)$ constraints
8:      $\boldsymbol{w} \leftarrow \text{PRUNER}(\mathcal{T}, \boldsymbol{w}^{(0)}, \mathcal{S}^{(t)})$
9:      # $\mathcal{O}(pB^2)$ constraints
10:     $\mathcal{S}^{\star} \leftarrow \text{ORACLE}(\mathcal{T}, \boldsymbol{w}^{(0)}, \boldsymbol{w}, s(\cdot), \tau(\alpha))$
11:     **if** $\mathcal{S}^{\star} \neq \varnothing$ **then**
12:         $\mathcal{S}^{(t+1)} \leftarrow \mathcal{S}^{(t)} \cup \mathcal{S}^{\star}$
13:         $t \leftarrow t + 1$
14:     **end if**
15: **until** $\mathcal{S}^{\star} = \varnothing$
16: **return** $\boldsymbol{w}$

---

$\tau(\alpha)$ using a dataset $\mathcal{D}_{\text{cal}}$; in our experiments, $\mathcal{D}_{\text{fit}}$ and $\mathcal{D}_{\text{cal}}$ are obtained by splitting the training data.

Algorithm 1 summarizes PINE. We maintain a growing constraint set $\mathcal{S}$ of inputs on which prediction equivalence is enforced during pruning. We warm-start with $\mathcal{S}^{(0)} \leftarrow \mathcal{D}_{\text{fit}}$. At each iteration $t$, given the current set $\mathcal{S}^{(t)}$, Pruner updates the weights to satisfy prediction equivalence on $\mathcal{S}^{(t)}$ (see Section 3.2); the Oracle then returns counterexamples $\mathcal{S}^{\star}$ within $\mathcal{X}_{\text{ID}}(\alpha)$, which we union into $\mathcal{S}^{(t+1)}$.

Since the plausible score $s(\boldsymbol{x})$ is included as a constraint

in the `Oracle` MILP, it should admit a corresponding tree-structured MILP encoding. Accordingly, the central questions in PINE can be summarized as follows:

*(i) How should we design a plausible score $s(\boldsymbol{x})$ that is easily embeddable in the MILP? (ii) How should we choose the threshold $\tau(\alpha)$ that separates in-/out-of-distribution inputs?*

In Section 4.1, we address (i) by adopting a Chow-Liu tree. In Section 4.2, we address (ii) by determining $\tau(\alpha)$ via the split conformal framework introduced in Section 3.3. Finally, in Section 4.3, we theoretically analyze how in-distribution prediction equivalence based on the Chow-Liu constraint affects compression and fidelity.

### 4.1. Chow-Liu Tree as a Plausible Score

We adopt the negative log-likelihood (NLL) of a Chow-Liu tree as the plausible score in PINE. Since PINE restricts the Oracle search region to the in-distribution region $\mathcal{X}_{\text{ID}}(\alpha)$ defined by $s(\boldsymbol{x}) \leq \tau(\alpha)$, it is desirable to use a score that captures the data distribution well while remaining computationally efficient and linearly embeddable in a MILP. Chow-Liu trees satisfy these requirements in terms of both distributional expressiveness and computational efficiency. More generally, PINE can incorporate tree-structured distribution constraints; in Section C we provide two alternative plausible scores and compare evaluation metrics.

A Chow-Liu tree (Chow & Liu, 1968) is a probabilistic model that approximates the dependency structure among features with a tree. It efficiently approximates the input distribution by maximizing the empirical likelihood within the class of tree-structured distributions. Equivalently, it is obtained by taking the maximum spanning tree over mutual information estimated from $\mathcal{D}_{\text{fit}}$.

We now define the Chow-Liu-based plausible score. First, we discretize each continuous feature into $B$ bins and denote the resulting discretized input vector by $\tilde{\boldsymbol{x}} = (\tilde{x}_1, \ldots, \tilde{x}_p) \in \{1, \ldots, B\}^p$, where $\tilde{x}_j$ is the bin index of feature $j$. The Chow-Liu tree factorizes the joint distribution of $\tilde{\boldsymbol{x}}$ as

$$p_{\mathrm{CL}}(\tilde{\boldsymbol{x}}) = p(\tilde{x}_r) \prod_{j \neq r} p(\tilde{x}_j \mid \tilde{x}_{\mathrm{pa}(j)}), \qquad (7)$$

where $r$ is an arbitrarily chosen root node and $\mathrm{pa}(j)$ denotes the parent of node $j$ in the rooted tree obtained by orienting edges away from $r$. We estimate $p(\tilde{x}_r)$ and $p(\tilde{x}_j \mid \tilde{x}_{\mathrm{pa}(j)})$ from $\mathcal{D}_{\mathrm{fit}}$, apply a small pseudo-count to handle unseen bins, and renormalize each table so that $p_{\mathrm{CL}}$ is a valid probability distribution. Based on this model, we define the plausible score of input $\boldsymbol{x}$ as

$$s(\boldsymbol{x}) := s_{\mathrm{CL}}(\tilde{\boldsymbol{x}}) := -\log p_{\mathrm{CL}}(\tilde{\boldsymbol{x}}). \qquad (8)$$

Smaller values of $s_{\mathrm{CL}}(\tilde{\boldsymbol{x}})$ correspond to higher likelihood, i.e., inputs that are more likely under the data distribution $P_0$. Therefore, the condition $s_{\mathrm{CL}}(\tilde{\boldsymbol{x}}) \leq \tau(\alpha)$ can be interpreted as a likelihood-based in-distribution constraint.

An advantage of Chow-Liu trees is that taking the negative log of Equation (7) decomposes the score in Equation (8) directly into a root-marginal term and conditional terms on the Chow-Liu tree edges:

$$s_{\mathrm{CL}}(\tilde{\boldsymbol{x}}) = \underbrace{-\log p(\tilde{x}_r)}_{\text{root-dependent term}} + \underbrace{\sum_{j \neq r} -\log p(\tilde{x}_j \mid \tilde{x}_{\mathrm{pa}(j)})}_{\text{edge-dependent terms}}. \quad (9)$$

This additive structure admits a compact MILP encoding of the in-distribution constraint $s_{\mathrm{CL}}(\tilde{\boldsymbol{x}}) \leq \tau(\alpha)$, requiring only a polynomial number of binary variables and linear constraints. In particular, with $p$ features and $B$ discretization bins per feature, the root term contributes a single-bin lookup and each edge term a parent-child bin-pair lookup, so the MILP encoding uses $\mathcal{O}(pB)$ binary variables to represent per-feature bin indicators and $\mathcal{O}(|E|B^2) = \mathcal{O}(pB^2)$ binary variables to represent bin pairs on edges. These counts are substantially smaller than the exponential size $\mathcal{O}(B^p)$ of the discretized input space, because the Oracle introduces bin-pair variables only for Chow-Liu edges rather than variables for all joint assignments.

We describe a concrete MILP encoding. Let $q_{i,b} \in \{0, 1\}$ indicate that feature $i$ falls into bin $b$, and enforce $\sum_{b=1}^{B} q_{i,b} = 1$ for each $i$. For each non-root node $j$ with parent $i = \mathrm{pa}(j)$, let $u_{i,j,b,b'} \in \{0, 1\}$ indicate the conjunction $q_{i,b} = 1$ and $q_{j,b'} = 1$. We encode this logical AND via

$$u_{i,j,b,b'} \leq q_{i,b}, \ u_{i,j,b,b'} \leq q_{j,b'}, \ u_{i,j,b,b'} \geq q_{i,b} + q_{j,b'} - 1. \qquad (10)$$

The root-marginal and edge-conditional terms in Equation (9) can be precomputed as coefficients for root-dependent bins and edge-dependent bin pairs, and $s_{\mathrm{CL}}(\tilde{\boldsymbol{x}})$ can be written as a linear sum of $q_{i,b}$ and $u_{i,j,b,b'}$.

In implementation, when discretizing a continuous feature $j$, we round the resulting bin boundaries on each feature axis to the set of split thresholds $\Theta_j$ used by the trained ensemble $\mathcal{T}$ (i.e., all split thresholds used for feature $j$). This rounding is necessary because the Oracle MILP encodes boundary comparisons for continuous feature $j$ using indicator variables defined on $\Theta_j$. If some bin boundaries do not lie in $\Theta_j$, the region defined by $s(\boldsymbol{x}) \leq \tau(\alpha)$ can differ from the region feasible in the MILP, and counterexamples in the intended in-distribution region may be missed. We provide the rounding procedure and complexity in Section C.

## 4.2. Calibration of the In-Distribution Region

We propose to use split conformal prediction to choose the threshold $\tau(\alpha)$ so that the in-distribution region $\mathcal{X}_{\mathrm{ID}}(\alpha)$ constructed by PINE contains future inputs (test data in our setting) with a desired coverage level $1 - \alpha$. In the following, we rely on the construction and computation described in Section 3.3 and summarize only the guarantees needed for PINE.

**Assumption 4.1** (Exchangeability). The inputs in the calibration set $\mathcal{D}_{\mathrm{cal}}$ and a future input $\boldsymbol{X}_{\mathrm{new}}$ follow the same data distribution $P_0$ and are exchangeable.

Under the exchangeability assumption, the split-conformal threshold $\tau(\alpha)$ provides a lower bound of $1 - \alpha$ on the probability that a future input lies in the in-distribution region:

$$\mathbb{P}[\boldsymbol{X}_{\mathrm{new}} \in \mathcal{X}_{\mathrm{ID}}(\alpha)] = \mathbb{P}[s(\boldsymbol{X}_{\mathrm{new}}) \leq \tau(\alpha)] \geq 1 - \alpha. \quad (11)$$

**Proposition 4.2** (Probabilistic predictive equivalence). *Under Assumption 4.1, the pruned model obtained by PINE provides a lower bound of $1 - \alpha$ on the probability of prediction equivalence for a future input. That is, the following bound holds:*

$$\mathbb{P}\left[\hat{y}(\boldsymbol{X}_{\mathrm{new}}; \boldsymbol{w}) = \hat{y}(\boldsymbol{X}_{\mathrm{new}}; \boldsymbol{w}^{(0)})\right] \geq 1 - \alpha. \qquad (12)$$

This guarantee is a direct consequence of the split conformal guarantee $\mathbb{P}[\boldsymbol{X}_{\mathrm{new}} \in \mathcal{X}_{\mathrm{ID}}(\alpha)] \geq 1 - \alpha$ together with exact prediction equivalence on $\mathcal{X}_{\mathrm{ID}}(\alpha)$. The probability in Equation (12) is marginal over the calibration set and the future input under exchangeability. Moreover, the guarantee assumes that each Oracle MILP is solved to certified optimality/infeasibility; with time limits, the absence of a found counterexample is not a certificate of equivalence.

## 4.3. Theoretical Analysis

In this section, we clarify how restricting the equivalence guarantee region from the entire input space $\mathcal{X}$ to the in-

distribution region $\mathcal{X}_{\mathrm{ID}}(\alpha) \subseteq \mathcal{X}$ affects compression and computational efficiency. We show that, when combined with the Chow-Liu distribution constraint, the number of discrete states that `Oracle` can explore is explicitly upper bounded in terms of $\tau(\alpha)$, and this upper bound changes exponentially through $\tau(\alpha)$.

First, since $\mathcal{X}_{\mathrm{ID}}(\alpha) \subseteq \mathcal{X}$, restricting guarantees to the in-distribution region relaxes constraints and thus enlarges the feasible set; therefore, the $L_0$ objective value $\|\boldsymbol{w}\|_0$ cannot worsen. In particular, when the solver finds an optimal solution, PINE cannot be worse than FIPE in terms of $\|\boldsymbol{w}\|_0$. Second, when requiring prediction equivalence over the entire input space $\mathcal{X}$ as in FIPE, the number of cells induced by the tree ensemble partition can grow rapidly in high dimensions. For example, let $\Theta_j$ denote the set of split thresholds used across the ensemble for feature $j \in \{1, \ldots, p\}$. Since axis $j$ is partitioned into at most $|\Theta_j| + 1$ intervals, the input space $\mathcal{X}$ is partitioned into at most $\prod_{j=1}^{p}(|\Theta_j| + 1)$ cells.

In contrast, under the Chow-Liu distribution constraint, writing the discretized input as $\tilde{\boldsymbol{x}} \in \{1, \ldots, B\}^p$, the `Oracle` search is restricted to the discrete state set

$$A_\tau := \{\tilde{\boldsymbol{x}} \mid -\log p_{\mathrm{CL}}(\tilde{\boldsymbol{x}}) \leq \tau\}. \tag{13}$$

The following proposition shows that, for any threshold $\tau > 0$, the cardinality of $A_\tau$ is upper bounded by $e^\tau$.

**Proposition 4.3** (Upper bound on in-distribution states under the Chow-Liu constraint). *Let $p_{\mathrm{CL}}(\tilde{\boldsymbol{x}})$ denote the Chow-Liu distribution defined in Section 4.1, viewed as a probability distribution over discretized inputs $\tilde{\boldsymbol{x}} \in \{1, \ldots, B\}^p$. For a threshold $\tau > 0$, define $A_\tau$ as in Equation* (13). *Then the number of elements in $A_\tau$ satisfies $|A_\tau| \leq e^\tau$.*

*Proof.* For any $\tilde{\boldsymbol{x}} \in A_\tau$, by definition we have $p_{\mathrm{CL}}(\tilde{\boldsymbol{x}}) \geq e^{-\tau}$. Since $p_{\mathrm{CL}}$ is a probability distribution, $1 \geq \sum_{\tilde{\boldsymbol{x}} \in A_\tau} p_{\mathrm{CL}}(\tilde{\boldsymbol{x}}) \geq |A_\tau| e^{-\tau}$, which implies $|A_\tau| \leq e^\tau$. $\square$

Therefore, setting $\tau = \tau(\alpha)$ yields $|A_{\tau(\alpha)}| \leq e^{\tau(\alpha)}$. Moreover, since the split-conformal threshold $\tau(\alpha)$ is monotone non-increasing in $\alpha$, increasing $\alpha$ monotonically non-increases the upper bound $e^{\tau(\alpha)}$ on the number of states that `Oracle` may need to search, potentially making counterexample search easier. Since the full discrete space has size $B^p$, an effective bound is $\min(B^p, e^{\tau(\alpha)})$; the Chow-Liu bound is most informative when $e^{\tau(\alpha)} \ll B^p$.

## 5. Experiments

We compare PINE with existing tree ensemble pruning methods on public tabular classification datasets. Our experiments are designed to answer the following research questions: (RQ1) Can PINE achieve a better fidelity-pruning trade-off than existing methods? (RQ2) Does the miscoverage level $\alpha$ consistently control the size of the in-distribution region $\mathcal{X}_{\mathrm{ID}}(\alpha)$ and the trade-off between pruning rate and fidelity? (RQ3) Do pruning methods without prediction equivalence guarantees still change predictions even when evaluation is restricted to the in-distribution region $\mathcal{X}_{\mathrm{ID}}(\alpha)$ defined by PINE?

### 5.1. Experimental Setup

**Datasets and splits.** As shown in Table 4, we use 12 public tabular classification datasets with varying numbers of samples, features, and classes, obtained from public benchmark sources including the UCI repository (Kelly et al.) and OpenML (Vanschoren et al., 2014). For each dataset, we split the data into three sets: a fit set $\mathcal{D}_{\mathrm{fit}}$ used to train the original ensemble and to estimate the score function $s(\cdot)$, a calibration set $\mathcal{D}_{\mathrm{cal}}$ used to calibrate the threshold, and a test set $\mathcal{D}_{\mathrm{test}}$ used only for final evaluation. To ensure a fair comparison, we use identical splits and random seeds for all methods and report results averaged over five seeds; details are provided in Section A. For the $\alpha$ sweep, we reuse the same calibration split for all $\alpha$ (fixed calibration seed) and only recompute $\tau(\alpha)$.

**Baselines.** We compare PINE with existing pruning baselines: FIPE and three PyPruning methods, IC, DREP, and MDEP. FIPE (Emine et al., 2025) is the faithful pruning baseline, which certifies prediction equivalence over the entire input space $\mathcal{X}$. Following the experimental setup of FIPE, we use IC (Lu et al., 2010), DREP (Li et al., 2012), and MDEP (Guo et al., 2018) from the PyPruning library (Buschjäger & Morik, 2021) as accuracy-oriented tree-level pruning baselines. These methods prune by specifying the number of trees retained after pruning. We further report results with ForestPrune (Liu & Mazumder, 2023) and the remaining PyPruning methods in Sections B.3 and B.5. We exclude LOP (Devos et al., 2025), whose structural decisions require a new method-specific Oracle constraint encoding, and Born-Again Tree Ensembles (Vidal & Schiffer, 2020), which distill the ensemble into a single tree and change the model class.

**Hyperparameters.** For the ensemble to be pruned, we follow prior work and train boosted tree ensembles using XGBoost (Chen & Guestrin, 2016). We fix the maximum depth to $D = 2$ and the number of trees to $M = 30$. For PINE, we sweep the miscoverage level $\alpha \in \{0.05, 0.1, 0.2, 0.4, 0.6, 0.8\}$ and calibrate the threshold $\tau(\alpha)$ on $\mathcal{D}_{\mathrm{cal}}$ via split conformal prediction for each $\alpha$. For the Chow-Liu-based score, we fix the number of discretization bins for continuous features to $B = 4$. Sensitivity to the number of bins is reported in Section B.6. FIPE has no additional hyperparameters. Since IC/DREP/MDEP

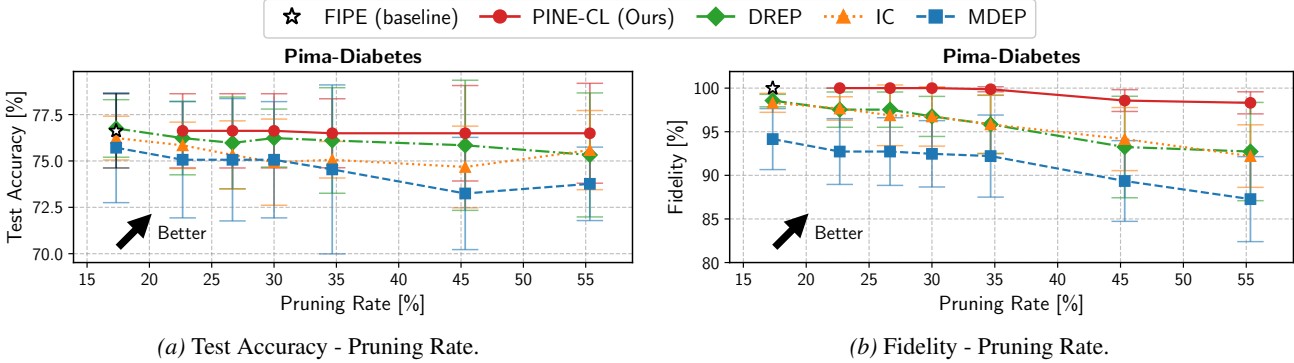

*(a)* Test Accuracy - Pruning Rate.          *(b)* Fidelity - Pruning Rate.

*Figure 3.* (RQ1) Comparison of test accuracy and fidelity on the Pima-Diabetes dataset. (a) Existing pruning methods, including FIPE, can maintain test accuracy comparable to the original model after pruning. (b) However, fidelity can differ substantially even when accuracies are similar. For existing methods, fidelity tends to decrease monotonically as the pruning rate increases, whereas PINE maintains both high pruning rates and high fidelity.

require a target number of trees, for each dataset and seed we set the target to the number of trees output by FIPE and by PINE (for each $\alpha$). Unless otherwise noted, we use default values for non-essential hyperparameters as summarized in Section A.

**Metrics.** We report test accuracy, fidelity, and pruning rate as evaluation metrics. The fidelity $\hat{\rho}$ on the test set $\mathcal{D}_{\text{test}}$ is defined as the fraction of inputs for which the pruned ensemble matches the original ensemble:

$$\hat{\rho} := \frac{1}{|\mathcal{D}_{\text{test}}|} \sum_{\boldsymbol{x} \in \mathcal{D}_{\text{test}}} \mathbb{1}\left[ \hat{y}(\boldsymbol{x}; \boldsymbol{w}) = \hat{y}(\boldsymbol{x}; \boldsymbol{w}^{(0)}) \right]. \quad (14)$$

We also define the pruning rate as the fraction of removed trees relative to the original number of trees $M$, i.e., $1 - \|\boldsymbol{w}\|_0 / M$; equivalently, the tree-count compression ratio is $M / \|\boldsymbol{w}\|_0$ (which we refer to as the compression ratio), where larger values indicate more compression. To validate the in-distribution region $\mathcal{X}_{\text{ID}}(\alpha)$ on which prediction equivalence is guaranteed, we additionally measure the empirical test coverage $\hat{\pi}_{\text{ID}}$ and the conditional fidelity $\hat{\rho}_{\text{ID}}$ on $\mathcal{X}_{\text{ID}}(\alpha)$:

$$\hat{\pi}_{\text{ID}} := \frac{1}{|\mathcal{D}_{\text{test}}|} \sum_{\boldsymbol{x} \in \mathcal{D}_{\text{test}}} \mathbb{1}[\boldsymbol{x} \in \mathcal{X}_{\text{ID}}(\alpha)] \quad (15)$$

$$\hat{\rho}_{\text{ID}} := \frac{\sum_{\boldsymbol{x} \in \mathcal{D}_{\text{test}}} \mathbb{1}[\boldsymbol{x} \in \mathcal{X}_{\text{ID}}(\alpha)] \, \mathbb{1}[\hat{y}(\boldsymbol{x}; \boldsymbol{w}) = \hat{y}(\boldsymbol{x}; \boldsymbol{w}^{(0)})]}{\sum_{\boldsymbol{x} \in \mathcal{D}_{\text{test}}} \mathbb{1}[\boldsymbol{x} \in \mathcal{X}_{\text{ID}}(\alpha)]}. \quad (16)$$

If split conformal calibration is appropriate, we expect $\hat{\pi}_{\text{ID}} \approx 1 - \alpha$, and if the prediction-equivalence guarantee succeeds we expect $\hat{\rho}_{\text{ID}} \approx 1$.

**Optimization.** Experiments are run on a machine with an Intel Core i7-11800H (8 cores / 16 threads), 2.3GHz CPU and 32GB RAM. For optimization, we use the commercial

solver `Gurobi v11.0.3`. All MILP solves in our main experiments terminated with optimality or with a certificate that no feasible solution exists (i.e., no counterexample) before the time limit, so the reported guarantees are certified. In this section, we report results for the $\|\boldsymbol{w}\|_0$ objective. In Section B.2, we additionally report results for an approximation that minimizes $\|\boldsymbol{w}\|_1$ to reduce computational cost and additional metrics such as runtime. Section B.7 also reports results on Random Forests and larger XGBoost ensembles ($M = 50$), showing similar trends.

**5.2. Results**

We organize the main results by research question. In particular, we focus on how test accuracy and fidelity behave as a function of pruning rate, and on whether the calibration of the in-distribution region $\mathcal{X}_{\text{ID}}(\alpha)$ remains valid on the test data $\mathcal{D}_{\text{test}}$.

**(RQ1) Fidelity vs. pruning rate.** In Figure 3, we visualize the relationship between pruning rate and (a) test accuracy and (b) fidelity. As shown in Figure 3a, existing pruning methods without prediction-equivalence guarantees (IC/DREP/MDEP) can maintain test accuracy comparable to the original model after pruning. However, Figure 3b shows that fidelity can differ substantially across methods even at the same pruning rate. These results highlight that similar accuracy does not necessarily imply prediction equivalence, which directly motivates model compression that preserves decision consistency.

FIPE, which guarantees equivalence over the entire input space, has fidelity 1 by definition, but its pruning rate can be limited because the counterexample search spans the entire input space. In contrast, PINE focuses the equivalence guarantee on the in-distribution region $\mathcal{X}_{\text{ID}}(\alpha)$, thereby maintaining high overall fidelity while achieving higher pruning

*Table 1.* Comparison of FIPE (baseline) and PINE-CL (ours) on Pima-Diabetes in terms of pruning rate PR (%), fidelity Fid. (%), runtime Time (s), and number of iterations Iter. (averaged over 5 seeds). Bold indicates the best value for each metric (higher is better for PR and Fid., lower is better for Time and Iter.).

| Method | $\alpha$ | PR ($\uparrow$) | Fid. ($\uparrow$) | Time ($\downarrow$) | Iter. ($\downarrow$) |
|---|---|---|---|---|---|
| FIPE | – | 17.3 | **100.0** | 42.5 | 24.6 |
| PINE-CL | 0.05 | 22.7 | **100.0** | 48.1 | 19.2 |
| | 0.1 | 26.7 | **100.0** | 48.0 | 19.6 |
| | 0.2 | 30.0 | **100.0** | 47.0 | 19.6 |
| | 0.4 | 34.7 | 99.9 | 33.8 | 15.2 |
| | 0.6 | 45.3 | 98.6 | 19.4 | 11.0 |
| | 0.8 | **55.3** | 98.3 | **12.0** | **7.8** |

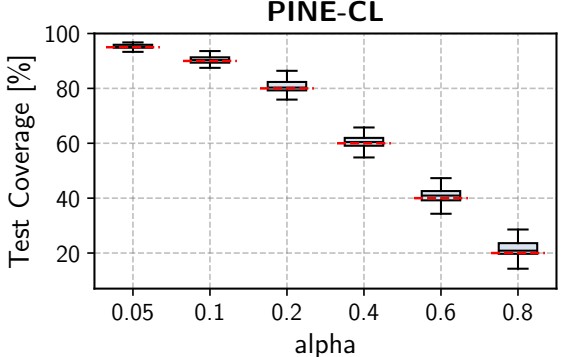

*Figure 4.* (RQ2) Empirical test coverage $\hat{\pi}_{\mathrm{ID}}$ on $\mathcal{D}_{\mathrm{test}}$ as a function of miscoverage $\alpha$. $\hat{\pi}_{\mathrm{ID}}$ closely tracks the target coverage $1 - \alpha$ (red dashed line), indicating that split-conformal threshold calibration $\tau(\alpha)$ is valid.

rates than FIPE. This trend is consistently observed across the 12 datasets: PINE increases mean pruning rate from 44.6% to 67.8% as $\alpha$ increases from 0.05 to 0.8, while mean fidelity remains between 99.96% and 99.15% (see Section B.1). Table 1 further reports pruning rate, fidelity $\hat{\rho}$, runtime, and number of iterations on Pima-Diabetes. These results demonstrate that PINE controls the trade-off between pruning rate and fidelity via $\alpha$.

**(RQ2) Controlling coverage with $\alpha$.** We examine whether the miscoverage level $\alpha$ effectively controls the coverage of the guarantee region. Figure 4 plots the empirical test coverage $\hat{\pi}_{\mathrm{ID}}$ as a function of $\alpha$. The figure shows that $\hat{\pi}_{\mathrm{ID}}$ closely tracks the target level $1 - \alpha$, indicating that split-conformal threshold calibration $\tau(\alpha)$ remains valid on real data. Therefore, by varying $\alpha$ we can explicitly adjust the size of $\mathcal{X}_{\mathrm{ID}}(\alpha)$, i.e., how broadly prediction equivalence is guaranteed.

**(RQ3) Fidelity within the in-distribution region.** We test whether pruning methods without prediction-equivalence guarantees still change predictions even when

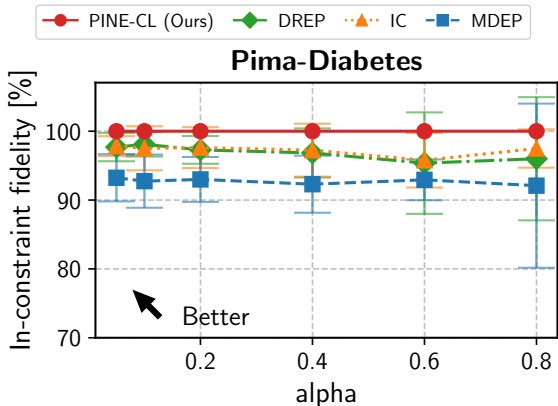

*Figure 5.* (RQ3) Conditional fidelity $\hat{\rho}_{\mathrm{ID}}$ within $\mathcal{X}_{\mathrm{ID}}(\alpha)$ on the Pima-Diabetes dataset. Although accuracy-oriented pruning methods (IC/DREP/MDEP) have higher fidelity when evaluation is restricted to the in-distribution region, they can still produce prediction disagreements.

restricted to the in-distribution region $\mathcal{X}_{\mathrm{ID}}(\alpha)$ defined by PINE. Concretely, we fix $\mathcal{X}_{\mathrm{ID}}(\alpha)$ constructed by PINE and evaluate, for each method, the conditional fidelity $\hat{\rho}_{\mathrm{ID}}$ defined in Equation (16). Observing $\hat{\rho}_{\mathrm{ID}} < 1$ implies that prediction changes occur even within the in-distribution region. Indeed, as shown in Figure 5, accuracy-oriented pruning methods (IC/DREP/MDEP) can produce prediction disagreements even within $\mathcal{X}_{\mathrm{ID}}(\alpha)$.

# 6. Discussion

## 6.1. Case Study

To explain why PINE achieves strong pruning performance, we analyze the counterexamples ignored by PINE on the Adult and COMPAS datasets. FIPE must preserve such counterexamples to guarantee prediction equivalence over the entire input space $\mathcal{X}$, whereas PINE can ignore them when they lie outside the calibrated region $\mathcal{X}_{\mathrm{ID}}(\alpha)$. Adult contains a point combining `Preschool` education with `13.5 years of education` and `Never-married` status with the `Wife` relationship role. COMPAS contains a point for which the Boolean features `prior offenses=0` and `prior offenses>3` are both true, a logically impossible combination. These examples are infeasible points; by not enforcing prediction equivalence on such points outside $\mathcal{X}_{\mathrm{ID}}(\alpha)$, PINE achieves higher compression while preserving decisions within $\mathcal{X}_{\mathrm{ID}}(\alpha)$.

## 6.2. Practical $\alpha$ Selection

By empirically selecting $\alpha$ post hoc on an additional data split, practitioners can obtain the most compressed PINE model that satisfies a target fidelity level. We introduce two selection rules for choosing $\alpha$ from a finite grid. The first is

an empirical selector, which follows the held-out selection principle used in LOP (Devos et al., 2025): among the candidate $\alpha$ values, it chooses the largest one whose fidelity on $\mathcal{D}_{\mathrm{sel}}$ meets the target fidelity. The second is a confidence-bound-based selector, which follows the finite-grid risk-control perspective of Learn-then-Test (Angelopoulos et al., 2025): among the candidate $\alpha$ values, it chooses the largest one whose Bonferroni-corrected Clopper-Pearson upper confidence bound on held-out mismatch risk is at most one minus the target fidelity; details are provided in Section B.4.

We evaluate these rules using a new four-way split that includes an additional selection set $\mathcal{D}_{\mathrm{sel}}$, separate from $\mathcal{D}_{\mathrm{fit}}$, $\mathcal{D}_{\mathrm{cal}}$, and $\mathcal{D}_{\mathrm{test}}$. We then select $\alpha$ according to whether the pruned model satisfies the target fidelity condition on $\mathcal{D}_{\mathrm{sel}}$. For a 95% fidelity target, the empirical selector chooses $\alpha = 0.95$ on all 12 datasets, achieving 70.8% mean pruning and 98.77% mean test fidelity. The confidence-bound-based selector is less aggressive, achieving 57.2% mean pruning and 99.72% mean test fidelity for the same target. Note that these test fidelities are empirical outcomes of the selection procedure, not additional conformal guarantees. Nevertheless, they show that practitioners can use a held-out selection split to choose $\alpha$ as a practical operating point under a target-fidelity requirement.

### 6.3. Scalability Analysis

To assess scalability of PINE, we conducted additional experiments over varying numbers of trees $M$ and maximum depths $D$. Specifically, we evaluate PINE at $\alpha = 0.8$ over $(M, D) \in \{10, 20, 30, 40, 50\} \times \{2, 3, 4, 5\}$. Table 2 fixes the number of trees at $M = 30$ and varies the maximum depth $D$, while Table 3 fixes the maximum depth at $D = 3$ and varies the number of trees $M$.

The results show that maximum depth $D$ is the key factor limiting scalable tree-level compression. In Table 2, increasing $D$ from 2 to 5 at $M = 30$ decreases the mean pruning rate by 32.50 pp, while mean fidelity remains near 99%. Over the same depth change, runtime grows from 3.82 s to 932.75 s and the number of iterations grows from 2.17 to 13.17. By contrast, at fixed $D = 3$, Table 3 shows that increasing $M$ from 10 to 50 keeps pruning substantial (39.17 to 51.11%) and fidelity near 99%, although runtime and iterations grow sharply. These results suggest that scalability is limited primarily by compression and runtime, rather than by a loss of fidelity. Deeper trees reduce attainable tree-level compression, while larger ensembles mainly make optimization substantially more expensive. This result can be interpreted as a consequence of deeper trees creating more localized decision regions, which makes more trees partially useful and harder to remove entirely under a prediction-equivalence constraint.

*Table 2.* Scaling with maximum depth $D$ for PINE over 12 datasets at $\alpha = 0.8$ and $M = 30$. PR$\leq 1\%$ counts the number of cases with pruning rate at most 1%.

| Depth | PR (%) | Fid. (%) | Time (s) | Iter. | PR$\leq 1\%$ |
|---|---|---|---|---|---|
| $D = 2$ | 66.94 | 99.25 | 3.82 | 2.17 | 0/12 |
| $D = 3$ | 51.11 | 99.50 | 21.77 | 5.58 | 1/12 |
| $D = 4$ | 38.33 | 98.87 | 180.36 | 11.33 | 3/12 |
| $D = 5$ | 34.44 | 99.32 | 932.75 | 13.17 | 6/12 |

*Table 3.* Scaling with the number of trees $M$ for PINE over 12 datasets at $\alpha = 0.8$ and $D = 3$. PR$\leq 1\%$ counts the number of cases with pruning rate at most 1%.

| Trees | PR (%) | Fid. (%) | Time (s) | Iter. | PR$\leq 1\%$ |
|---|---|---|---|---|---|
| $M = 10$ | 39.17 | 99.80 | 1.89 | 1.42 | 0/12 |
| $M = 20$ | 47.92 | 99.36 | 10.67 | 3.50 | 1/12 |
| $M = 30$ | 51.11 | 99.50 | 21.77 | 5.58 | 1/12 |
| $M = 40$ | 50.42 | 99.24 | 355.33 | 9.25 | 1/12 |
| $M = 50$ | 47.00 | 99.27 | 755.46 | 15.83 | 0/12 |

## 7. Conclusion

This work addresses achieving high pruning rates and high fidelity for decision tree ensembles and proposes PINE, which guarantees prediction equivalence on an in-distribution region. PINE calibrates this region via split conformal prediction and prunes so predictions match on it. Under the exchangeability assumption, PINE provides a probabilistic prediction-equivalence guarantee: with probability at least $1 - \alpha$, predictions on a future input match those of the original model.

In experiments, accuracy-oriented pruning methods can maintain test accuracy while fidelity varies substantially. PINE maintains high fidelity and achieves higher pruning rates than faithful pruning methods that enforce equivalence over the entire input space by avoiding restrictive out-of-distribution constraints.

One limitation is that our guarantee relies on the validity of split conformal calibration; under distribution shift, the guarantee may weaken. Addressing this will require combining recalibration when distribution shift is detected (Tibshirani et al., 2019; Gibbs & Candès, 2021) with mechanisms to monitor the guarantee region. Moreover, because our framework solves counterexample search as a MILP, computational cost can increase, especially in multi-class settings; thus, improving scalability to larger ensembles is an important direction. Overall, improving robustness via recalibration and monitoring and enhancing computational efficiency via better Oracle design are key steps toward broader deployment of pruning methods with guarantees.

## Impact Statement

Our work aims to make post-hoc compression of tree ensembles more reliable by providing verifiable prediction-equivalence guarantees on a calibrated in-distribution region. The guarantee can reduce inference cost while preserving decision consistency in settings where model behavior must be audited or verified. However, the guarantee relies on exchangeability and on solving the underlying MILPs to certified optimality; under distribution shift or solver time limits, the guarantee may weaken and could be misinterpreted as an unconditional safety guarantee beyond the stated assumptions. We encourage practitioners to monitor for distribution shift, report solver statuses, and validate guarantees in the intended deployment setting.

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

# A. Experimental Details

We summarize the experimental details described in Section 5.

*Table 4.* Dataset statistics used in the experiments.

| Dataset | #Samples | #Features | #Classes |
|---|---|---|---|
| Adult | 32561 | 14 | 2 |
| Balance-Scale | 625 | 4 | 3 |
| Breast-Cancer-Wisconsin | 683 | 9 | 2 |
| COMPAS-ProPublica | 6907 | 12 | 2 |
| ELEC2 | 38474 | 7 | 2 |
| FICO | 10459 | 17 | 2 |
| HTRU2 | 17898 | 8 | 2 |
| JM1 | 10885 | 21 | 2 |
| Pima-Diabetes | 768 | 8 | 2 |
| PoL | 10082 | 26 | 2 |
| Seeds | 210 | 7 | 3 |
| Spambase | 4601 | 57 | 2 |

**Data splits.** We use a three-way split: 64% of the full data is used as the fit set $\mathcal{D}_{\text{fit}}$, 16% as the calibration set $\mathcal{D}_{\text{cal}}$, and 20% as the test set $\mathcal{D}_{\text{test}}$. Thus, the final fit/calibration/test proportions are $64:16:20$. We reuse the same split across methods and report results averaged over five seeds.

**MILP configuration.** We use `Gurobi` `v11.0.3` with a per-call time limit of 120 seconds, one thread, and at most 10,000 Oracle calls. Theoretical guarantees apply when the MILPs are solved to certified optimality/infeasibility.

**Plausible-score hyperparameters.** Unless stated otherwise, Chow-Liu and Leaf Support use Laplace smoothing $\beta = 1.0$; Isolation Forest uses $K = 30$ trees with `max_samples = 256` (see Section C for Leaf Support/Isolation Forest definitions).

**Code Availability.** The implementation and experiment scripts are available at `https://github.com/Haruk1y/pine`.

**Baseline implementations.** For baseline implementations, we used the public repositories for FIPE (`https://github.com/eminyous/fipe`), PyPruning (`https://github.com/sbuschjaeger/PyPruning`), and ForestPrune (`https://github.com/mazumder-lab/ForestPrune`).

# B. Detailed Experimental Results

## B.1. Results for $L_0$ Minimization

For the $\|\boldsymbol{w}\|_0$ objective, we show the trade-off curves over the 12 datasets in the following panels. Table 5 summarizes the per-dataset pruning rate, fidelity, and runtime.

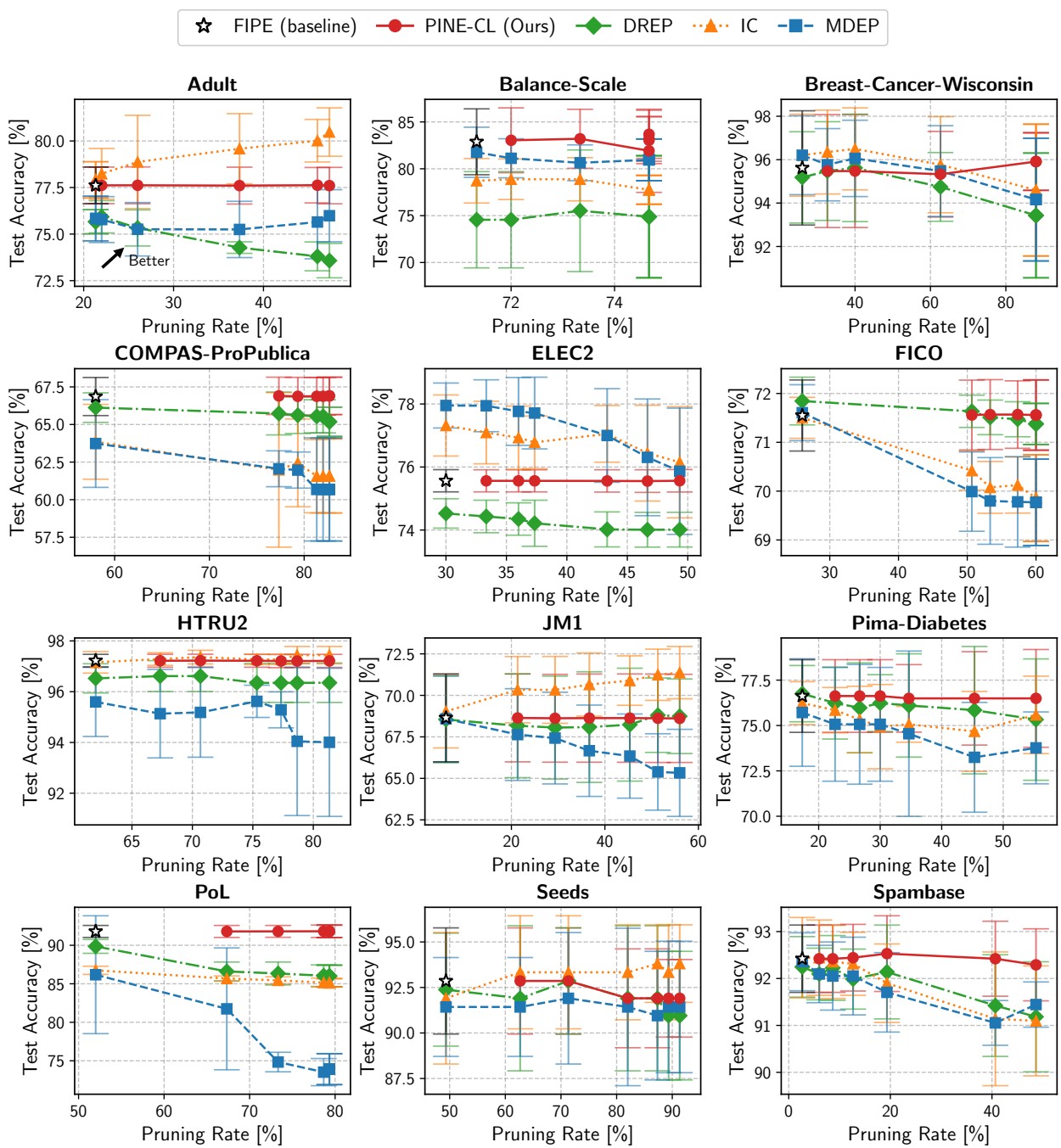

*Figure 6.* ($L_0$) Test Accuracy - Pruning Rate (12 datasets).

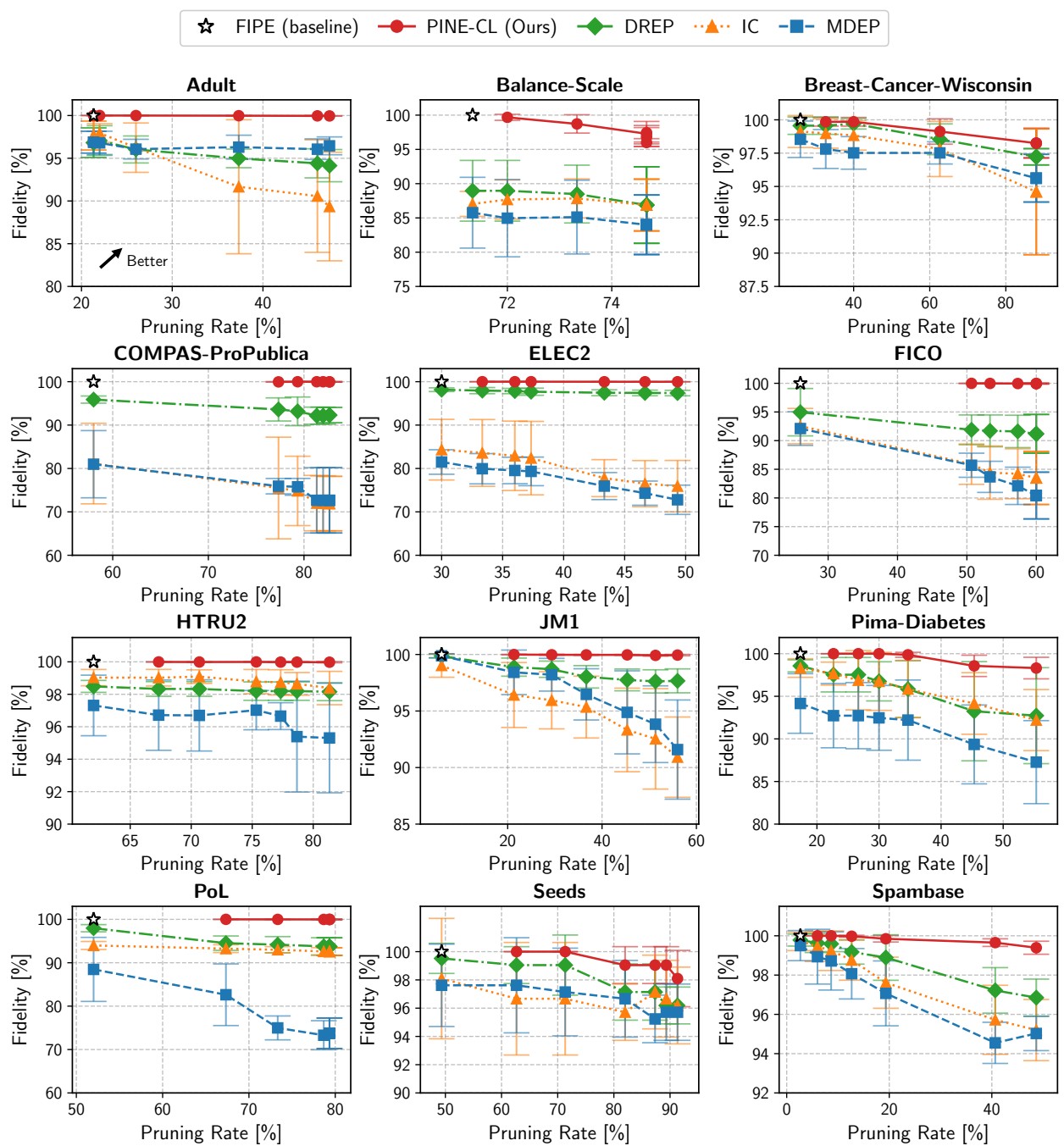

*Figure 7.* ($L_0$) Fidelity - Pruning Rate (12 datasets).

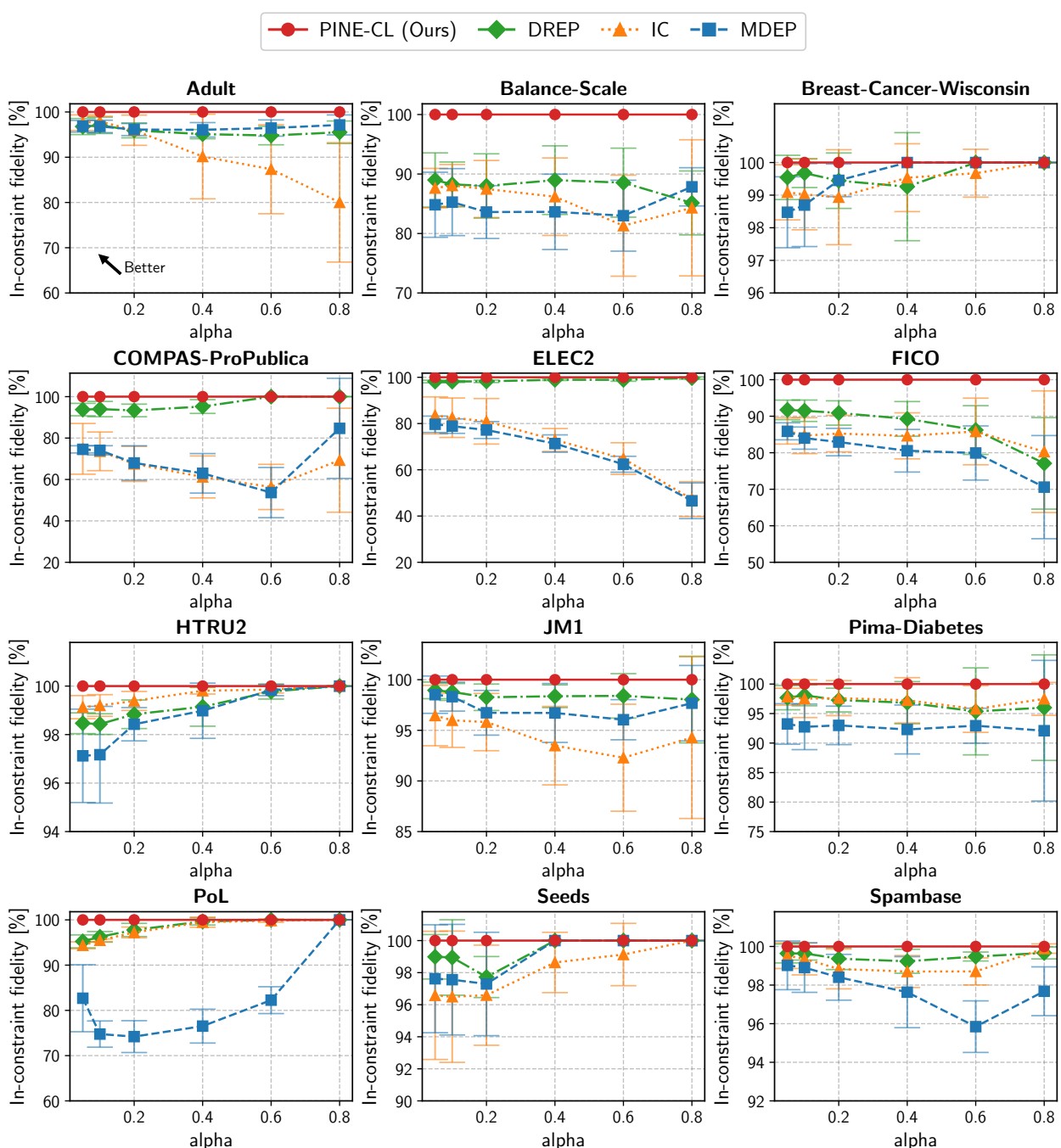

*Figure 8.* ($L_0$) In-constraint Fidelity - alpha (12 datasets).

*Table 5.* Per-dataset pruning rate, fidelity, and runtime for FIPE (Baseline) and PINE-CL (Ours) across $\alpha$ (mean $\pm$ std over 5 seeds). Best values in each row are boldfaced. PINE columns correspond to $\alpha \in \{0.05, 0.1, 0.2, 0.4, 0.6, 0.8\}$. Dataset rows use short names (Balance=Balance-Scale, Breast=Breast-Cancer-Wisconsin, COMPAS=COMPAS-ProPublica, Pima=Pima-Diabetes).

| Dataset | Metric | FIPE (Baseline) | PINE-CL (Ours) | | | | | |
| | | | $\alpha = 0.05$ | $\alpha = 0.1$ | $\alpha = 0.2$ | $\alpha = 0.4$ | $\alpha = 0.6$ | $\alpha = 0.8$ |
| --- | --- | --- | --- | --- | --- | --- | --- | --- |
| Adult | Pruning Rate | 21.3±3.4 | 21.3±3.4 | 22.0±4.5 | 26.0±4.9 | 37.3±6.5 | 46.0±3.9 | **47.3±3.9** |
| | Fidelity | **100.0±0.0** | **100.0±0.0** | **100.0±0.0** | **100.0±0.0** | **100.0±0.0** | **100.0±0.0** | **100.0±0.0** |
| | Time | 74.7±17.6 | 252.8±37.6 | 242.0±37.9 | 185.2±13.9 | 106.2±30.2 | 31.9±10.0 | **9.3±3.5** |
| Balance | Pruning Rate | 71.3±2.7 | 72.0±2.7 | 73.3±2.1 | **74.7±2.7** | **74.7±2.7** | **74.7±2.7** | **74.7±2.7** |
| | Fidelity | **100.0±0.0** | 99.7±0.4 | 98.7±1.2 | 97.3±1.1 | 97.4±1.5 | 96.8±1.2 | 96.0±0.5 |
| | Time | 43.2±15.2 | 51.3±23.1 | 39.5±27.2 | 27.7±16.1 | 16.4±6.7 | 13.5±5.5 | **11.7±5.5** |
| Breast | Pruning Rate | 26.0±6.5 | 32.7±6.8 | 40.0±10.5 | 62.7±18.3 | **88.0±1.6** | **88.0±1.6** | **88.0±1.6** |
| | Fidelity | **100.0±0.0** | 99.9±0.3 | 99.9±0.3 | 99.1±0.9 | 98.2±1.0 | 98.2±1.0 | 98.2±1.0 |
| | Time | 45.2±11.7 | 44.0±11.2 | 34.0±11.7 | 21.4±15.2 | 0.7±0.2 | **0.5±0.2** | **0.5±0.2** |
| COMPAS | Pruning Rate | 58.0±4.5 | 77.3±1.3 | 79.3±1.3 | 81.3±1.6 | 82.0±1.6 | **82.7±2.5** | **82.7±2.5** |
| | Fidelity | **100.0±0.0** | **100.0±0.1** | **100.0±0.0** | **100.0±0.0** | **100.0±0.0** | **100.0±0.1** | **100.0±0.1** |
| | Time | 12.2±2.0 | 1.7±0.3 | 1.4±0.6 | 0.6±0.3 | 0.5±0.1 | **0.3±0.0** | **0.3±0.0** |
| ELEC2 | Pruning Rate | 30.0±3.7 | 33.3±6.7 | 36.0±6.5 | 37.3±8.0 | 43.3±7.0 | 46.7±4.7 | **49.3±3.3** |
| | Fidelity | **100.0±0.0** | **100.0±0.0** | **100.0±0.0** | **100.0±0.0** | **100.0±0.0** | **100.0±0.0** | **100.0±0.0** |
| | Time | 45.2±11.1 | 39.6±14.3 | 35.1±11.7 | 25.6±9.5 | 16.6±9.5 | 7.0±1.5 | **2.8±0.5** |
| FICO | Pruning Rate | 26.0±3.9 | 50.7±3.9 | 53.3±4.2 | 57.3±3.3 | **60.0±3.0** | **60.0±3.0** | **60.0±3.0** |
| | Fidelity | **100.0±0.0** | **100.0±0.0** | **100.0±0.0** | **100.0±0.0** | **100.0±0.1** | **100.0±0.1** | **100.0±0.1** |
| | Time | 96.1±17.2 | 30.3±8.6 | 28.8±10.2 | 12.8±7.1 | 5.0±3.2 | **3.2±0.2** | 3.3±0.2 |
| HTRU2 | Pruning Rate | 62.0±11.7 | 67.3±10.6 | 70.7±10.4 | 75.3±10.0 | 77.3±6.5 | 78.7±6.9 | **81.3±3.4** |
| | Fidelity | **100.0±0.0** | **100.0±0.0** | **100.0±0.0** | **100.0±0.0** | **100.0±0.0** | **100.0±0.0** | **100.0±0.0** |
| | Time | 4.7±2.1 | 4.1±2.2 | 2.5±1.6 | 1.5±1.2 | 1.5±1.2 | 0.9±0.7 | **0.5±0.0** |
| JM1 | Pruning Rate | 6.0±5.3 | 21.3±10.9 | 29.3±7.7 | 36.7±8.2 | 45.3±5.0 | 51.3±5.8 | **56.0±6.5** |
| | Fidelity | **100.0±0.0** | **100.0±0.0** | **100.0±0.0** | **100.0±0.0** | **100.0±0.0** | 99.9±0.1 | 99.9±0.1 |
| | Time | 165.5±59.6 | 68.0±37.3 | 51.6±28.5 | 34.6±17.5 | 13.3±3.5 | 7.6±2.5 | **3.4±2.6** |
| Pima | Pruning Rate | 17.3±8.3 | 22.7±10.6 | 26.7±13.5 | 30.0±15.2 | 34.7±17.1 | 45.3±19.4 | **55.3±17.7** |
| | Fidelity | **100.0±0.0** | **100.0±0.0** | **100.0±0.0** | **100.0±0.0** | 99.9±0.3 | 98.6±1.1 | 98.3±1.1 |
| | Time | 42.5±17.8 | 48.1±26.6 | 48.0±28.6 | 47.0±28.4 | 33.8±16.9 | 19.4±10.2 | **12.0±9.9** |
| PoL | Pruning Rate | 52.0±6.9 | 67.3±10.6 | 73.3±8.7 | 78.7±5.0 | **79.3±5.3** | **79.3±5.3** | **79.3±5.3** |
| | Fidelity | **100.0±0.0** | **100.0±0.0** | **100.0±0.0** | **100.0±0.0** | **100.0±0.0** | **100.0±0.0** | **100.0±0.0** |
| | Time | 12.6±4.3 | 5.9±2.4 | 3.1±1.7 | 0.7±0.1 | 0.6±0.1 | 0.6±0.1 | **0.5±0.1** |
| Seeds | Pruning Rate | 49.3±12.5 | 62.7±10.0 | 71.3±9.6 | 82.0±6.2 | 87.3±4.9 | 89.3±3.3 | **91.3±1.6** |
| | Fidelity | **100.0±0.0** | **100.0±0.0** | **100.0±0.0** | 99.0±1.2 | 99.0±1.2 | 99.0±1.2 | 98.1±1.8 |
| | Time | 42.9±17.2 | 50.7±20.2 | 30.9±16.4 | 17.9±16.1 | 3.3±2.5 | 1.7±0.7 | **1.0±0.5** |
| Spambase | Pruning Rate | 2.7±3.9 | 6.0±6.5 | 8.7±6.9 | 12.7±8.0 | 19.3±7.4 | 40.7±7.1 | **48.7±4.5** |
| | Fidelity | **100.0±0.0** | **100.0±0.0** | **100.0±0.0** | **100.0±0.0** | 99.8±0.1 | 99.7±0.2 | 99.4±0.3 |
| | Time | 875.5±339.6 | 272.9±50.5 | 217.1±35.9 | 114.5±30.7 | 49.2±7.3 | 19.0±5.3 | **6.2±2.4** |

## B.2. Results for $L_1$ Minimization

We report results when approximating the $\|\boldsymbol{w}\|_0$ objective by minimizing $\|\boldsymbol{w}\|_1$ instead. Compared to $\|\boldsymbol{w}\|_0$, the MILP can be easier to solve, but the sparsity of the solution may be affected by the approximation.

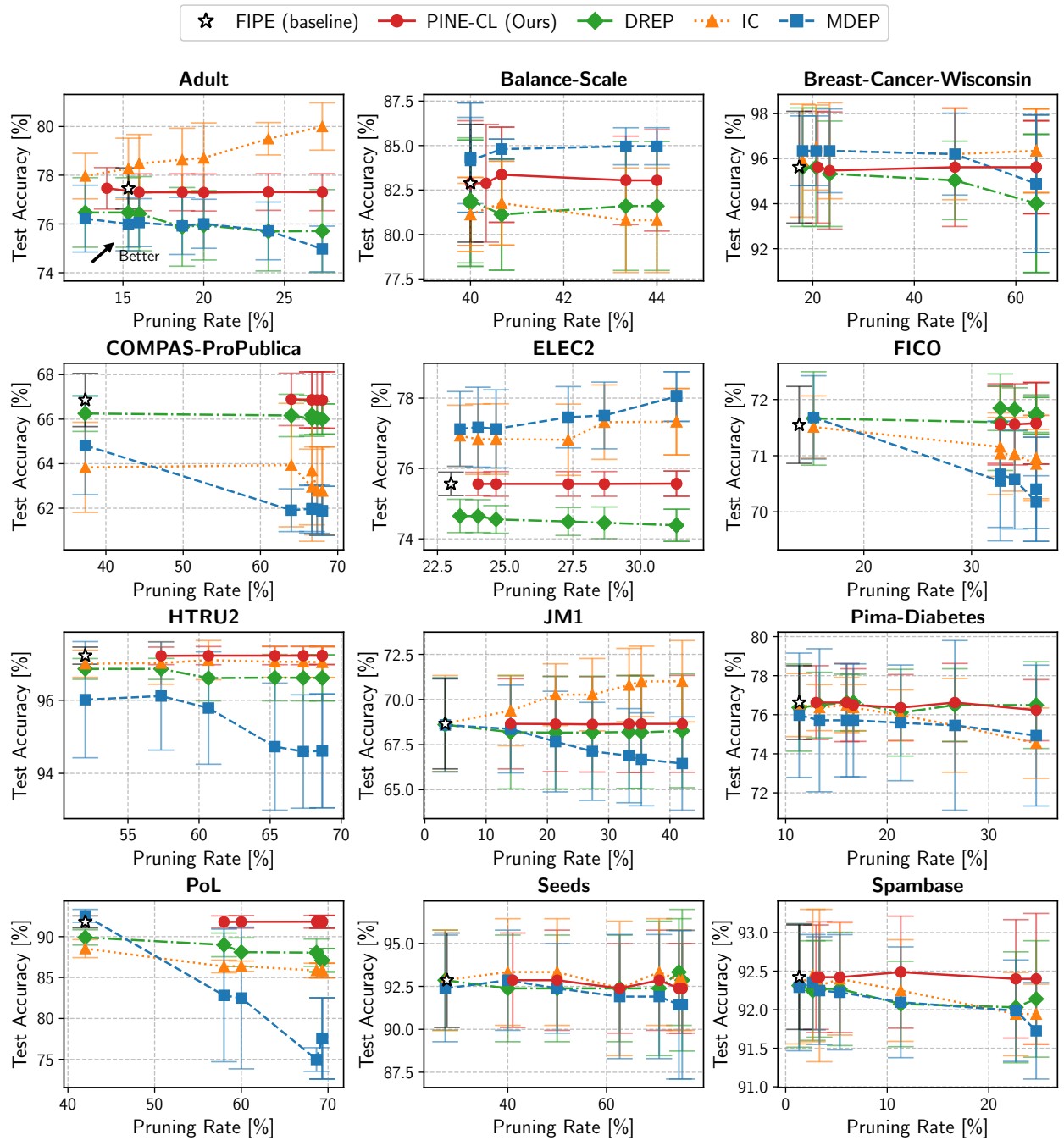

Figure 9. ($L_1$) Test Accuracy - Pruning Rate (12 datasets).

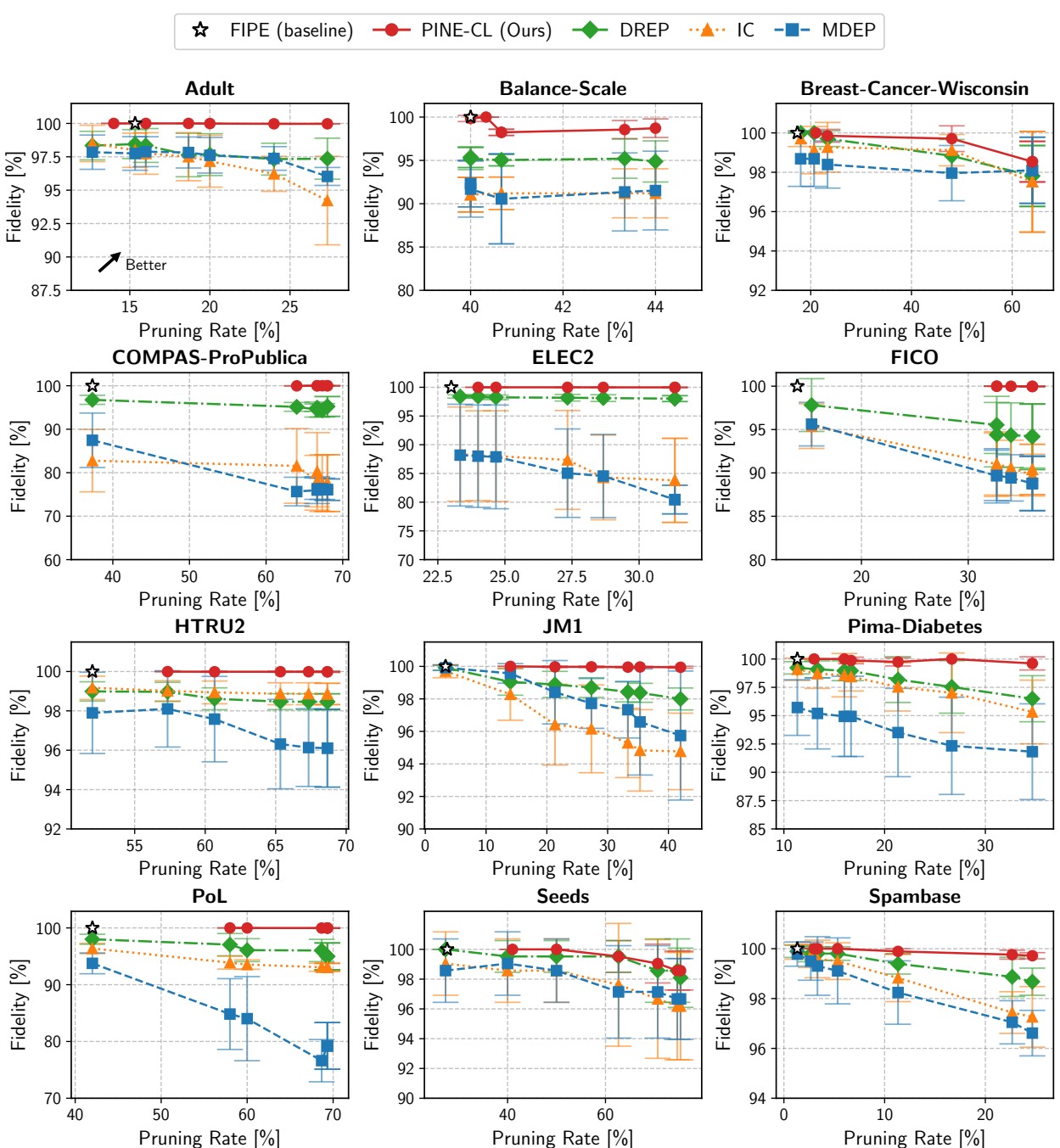

*Figure 10.* ($L_1$) Fidelity - Pruning Rate (12 datasets).

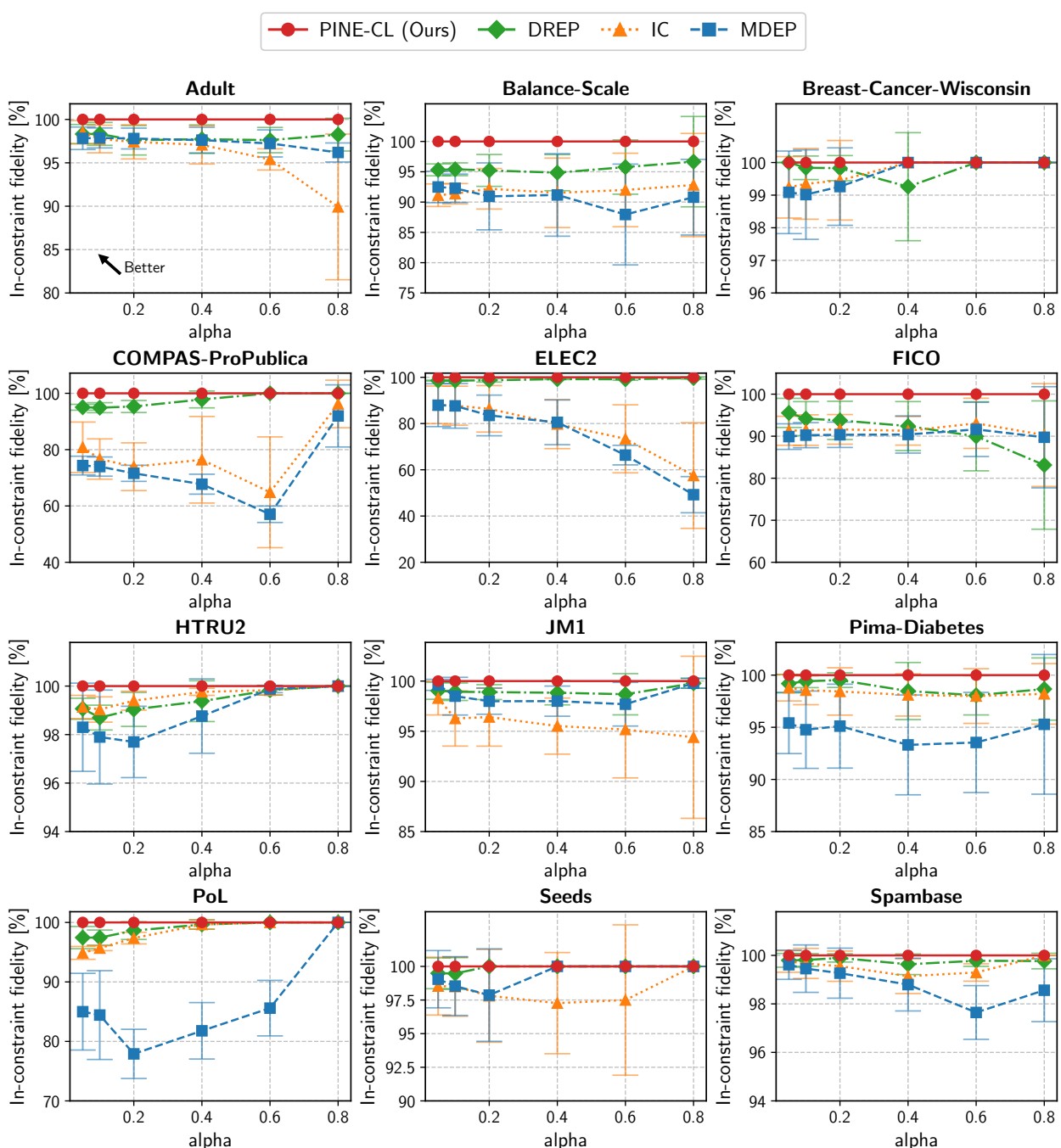

*Figure 11.* ($L_1$) In-constraint Fidelity - alpha (12 datasets).

## B.3. Comparison Against Additional Pruning Baselines

We compare PINE with two additional families of pruning baselines to test whether its fidelity advantage is robust across broader pruning methods. The first family contains all 15 PyPruning baselines, which are accuracy-oriented methods that select a target number of retained trees. The second family contains three simple heuristics: `LeafMagnitude-TopK` scores a tree by its average absolute leaf value, `TreeGain-TopK` scores a tree by its total split gain, and `FeatImpTree-TopK` scores a tree by summing feature-importance-weighted split contributions. We evaluate all methods on 12 datasets over five random seeds using XGBoost ensembles with $M = 30$ trees and maximum depth $D = 2$; for each dataset, seed, and $\alpha$, each baseline is given the number of trees retained by PINE as its target ensemble size.

Table 6 reports the mean fidelity across datasets and seeds for all methods. PINE attains the highest mean fidelity, and even the strongest non-PINE baseline, `cluster_centroids`, remains 2.83 pp lower. Among the simple Top-K heuristics, `LeafMagnitude-TopK` and `TreeGain-TopK` rank near the top of the non-PINE baselines, but they are still lower than PINE by 3.60 and 4.07 pp, respectively. This suggests that simpler heuristic methods can identify useful trees, but may not be sufficient to preserve the original ensemble's decisions as well as PINE.

*Table 6.* Fidelity comparison against additional pruning baselines.

| Method | Fid. [%] |
|---|---|
| PINE-CL | **99.57** |
| cluster_centroids | 96.74 |
| LeafMagnitude-TopK | 95.97 |
| TreeGain-TopK | 95.50 |
| error_ambiguity | 95.42 |
| drep | 95.41 |
| individual_error | 95.23 |
| FeatImpTree-TopK | 94.41 |
| largest_mean_distance | 94.31 |
| reduced_error | 94.22 |
| individual_kappa_statistic | 92.84 |
| complementariness | 92.75 |
| cluster_accuracy | 91.07 |
| individual_contribution | 90.84 |
| combined_error | 89.69 |
| margin_distance | 88.82 |
| individual_margin_diversity | 88.50 |
| combined | 86.90 |
| reference_vector | 86.59 |

## B.4. Practical $\alpha$ Selection

We describe two held-out $\alpha$-selection rules that turn the finite grid of calibrated PINE models into a practical operating-point choice. Let $\mathcal{A} \subset (0, 1)$ be the pre-specified finite set of candidate $\alpha$ values. Let $\rho_\star \in (0, 1)$ denote the target fidelity, and let $\mathcal{D}_{\text{sel}} = \{\boldsymbol{x}_i^{\text{sel}}\}_{i=1}^{n_{\text{sel}}} \subseteq \mathcal{X}$ be the selection split with $n_{\text{sel}} \in \mathbb{N}$. For each $\alpha \in \mathcal{A}$, let $\boldsymbol{w}_\alpha \in \mathbb{R}_{\geq 0}^M$ denote the PINE weights obtained at that candidate, and define the held-out mismatch rate $\hat{r}_{\text{sel}}(\alpha) \in [0, 1]$ as the fraction of inputs in $\mathcal{D}_{\text{sel}}$ on which PINE and the original ensemble predict different labels:

$$\hat{r}_{\text{sel}}(\alpha) = \frac{1}{n_{\text{sel}}} \sum_{\boldsymbol{x} \in \mathcal{D}_{\text{sel}}} \mathbb{1}\left[\hat{y}(\boldsymbol{x}; \boldsymbol{w}_\alpha) \neq \hat{y}(\boldsymbol{x}; \boldsymbol{w}^{(0)})\right].$$

Let $K_{\text{sel}}(\alpha) \in \{0, \ldots, n_{\text{sel}}\}$ denote the numerator in the above expression, i.e., the number of inputs in $\mathcal{D}_{\text{sel}}$ on which the two predictions differ, so that $\hat{r}_{\text{sel}}(\alpha) = K_{\text{sel}}(\alpha)/n_{\text{sel}}$. The empirical selector chooses the largest $\alpha \in \mathcal{A}$ satisfying $1 - \hat{r}_{\text{sel}}(\alpha) \geq \rho_\star$. The confidence-bound-based selector is more conservative and uses a Bonferroni finite-grid risk-control rule in the spirit of Learn-then-Test (Angelopoulos et al., 2025): with failure probability $\delta = 0.05 \in (0, 1)$, it chooses the largest $\alpha \in \mathcal{A}$ satisfying

$$U_{\text{CP}}(K_{\text{sel}}(\alpha), n_{\text{sel}}, \delta/|\mathcal{A}|) \leq 1 - \rho_\star. \tag{17}$$

Here, $U_{\text{CP}}(k, n, \eta)$, with $k \in \{0, \ldots, n\}$, $n \in \mathbb{N}$, and $\eta \in (0, 1)$, is the one-sided Clopper-Pearson upper confidence bound on a binomial mismatch risk after observing $k$ prediction mismatches among $n$ selection examples. For a pre-specified finite

grid and an independent selection split, the Bonferroni correction gives simultaneous control of the population mismatch risks over all candidates with probability at least $1 - \delta$; therefore any data-dependent choice among the certified candidates, including the largest $\alpha$, is covered by this risk-control statement. If no candidate $\alpha$ satisfies the confidence-bound criterion, we fall back to the unpruned ensemble.

We evaluate these selectors with a fit/calibration/selection/test split with proportions $48\!:\!16\!:\!16\!:\!20$ and the candidate grid $\mathcal{A} = \{0.05, 0.1, 0.2, 0.4, 0.6, 0.8, 0.9, 0.95\}$. The selection split is not used to fit the original ensemble, calibrate $\tau(\alpha)$, or construct the candidate PINE models, and $\mathcal{D}_{\text{test}}$ is used only for final evaluation. Table 7 shows that the empirical selector is more aggressive, while the confidence-bound-based selector trades pruning for higher final fidelity. For example, at the 95% target, the empirical selector chooses $\alpha = 0.95$ on all 12 datasets and achieves 70.8% mean pruning with 98.77% mean test fidelity, whereas the confidence-bound-based selector achieves 57.2% mean pruning with 99.72% mean test fidelity.

*Table 7.* Held-out fidelity-targeted $\alpha$ selection over 12 datasets. FB denotes fallback to the unpruned ensemble.

| Target | Selector | Selected $\alpha$ and number of FB across datasets | PR [%] | Fid. [%] |
|---|---|---|---|---|
| 95% | Empirical | $\alpha = 0.95$ on 12/12 | 70.8 | 98.77 |
| 95% | Confidence-bound | $\alpha = 0.95$ on 8/12, $\alpha = 0.4$ on 1/12, $\alpha = 0.2$ on 2/12, FB on 1/12 | 57.2 | 99.72 |
| 99% | Empirical | $\alpha = 0.95$ on 8/12, $\alpha = 0.6$ on 1/12, $\alpha = 0.4$ on 1/12, $\alpha = 0.2$ on 2/12 | 61.1 | 98.76 |
| 99% | Confidence-bound | $\alpha = 0.95$ on 7/12, $\alpha = 0.4$ on 1/12, FB on 4/12 | 39.2 | 99.96 |

## B.5. Comparison Against ForestPrune

We compare PINE with ForestPrune (Liu & Mazumder, 2023) to clarify how the empirical trade-off changes when the pruning method targets leaf-level size reduction under an accuracy constraint rather than prediction equivalence. ForestPrune prunes tree ensembles by making per-tree contiguous depth-cut decisions and selecting a smaller leaf-level representation subject to an allowed validation-accuracy drop. In contrast, PINE keeps the tree structures fixed, removes or reweights whole trees, and explicitly preserves the original ensemble's predictions on the calibrated in-distribution region.

We evaluate ForestPrune post hoc on the 10 binary datasets for which the ForestPrune pipeline applies, using the same data splits as PINE. ForestPrune uses the conservative setting `abserr=0.005`, where `abserr` bounds the allowed validation-accuracy drop relative to the original ensemble. In this comparison, ID fidelity (ID Fid.) denotes, for both methods, the conditional fidelity $\hat{\rho}_{\text{ID}}$ in Equation (16), evaluated on PINE's calibrated in-distribution region $\mathcal{X}_{\text{ID}}(0.8)$. Table 8 shows the main trade-off. At depth 2, PINE has both higher leaf pruning and higher fidelity. At depth 4, ForestPrune prunes many more leaves and runs faster, but its overall fidelity drops more noticeably.

*Table 8.* Aggregate post-hoc comparison with ForestPrune on 10 binary datasets. ForestPrune uses `abserr=0.005`; PINE uses $\alpha = 0.8$.

| Setting | ForestPrune | | | | PINE | | | |
|---|---|---|---|---|---|---|---|---|
| | Leaf PR [%] | Fid. [%] | ID Fid. [%] | Time [s] | Leaf PR [%] | Fid. [%] | ID Fid. [%] | Time [s] |
| $(M, D) = (30, 2)$ | 58.05 | 96.65 | 99.24 | 8.09 | 63.64 | 99.42 | 100.00 | 3.91 |
| $(M, D) = (30, 4)$ | 74.22 | 95.56 | 98.11 | 12.07 | 29.23 | 99.92 | 100.00 | 200.34 |

We further report dataset-level results at depth $D = 2$ to show that the aggregate ForestPrune trade-off is not driven by a single dataset. ForestPrune sometimes removes more leaves than PINE, but its overall fidelity can be substantially lower; PINE keeps ID fidelity at 100% in all rows of this slice because it explicitly certifies equivalence on $\mathcal{X}_{\text{ID}}(0.8)$. These results support the intended use distinction: ForestPrune is preferable when aggressive leaf-level compression under an accuracy bound is the main goal, whereas PINE is preferable when preserving the deployed ensemble's decisions on likely future inputs is the main goal.

*Table 9.* Dataset-level ForestPrune comparison for $(M, D) = (30, 2)$.

| | ForestPrune | | | PINE | | |
|---|---|---|---|---|---|---|
| Dataset | Leaf PR [%] | Fid. [%] | ID Fid. [%] | Leaf PR [%] | Fid. [%] | ID Fid. [%] |
| Adult | 58.3 | 96.02 | 94.02 | 43.3 | 99.97 | 100.00 |
| Breast | 27.1 | 100.00 | 100.00 | 86.4 | 98.54 | 100.00 |
| COMPAS | 98.3 | 90.09 | 100.00 | 83.3 | 99.93 | 100.00 |
| ELEC2 | 28.3 | 96.14 | 100.00 | 53.3 | 99.99 | 100.00 |
| FICO | 13.3 | 98.52 | 99.66 | 56.7 | 99.86 | 100.00 |
| HTRU2 | 98.3 | 99.78 | 100.00 | 86.7 | 100.00 | 100.00 |
| JM1 | 75.0 | 96.46 | 98.72 | 53.3 | 100.00 | 100.00 |
| Pima | 96.7 | 92.21 | 100.00 | 53.3 | 96.75 | 100.00 |
| PoL | 66.7 | 98.41 | 100.00 | 70.0 | 100.00 | 100.00 |
| Spambase | 18.3 | 98.91 | 100.00 | 50.0 | 99.13 | 100.00 |

## B.6. Sensitivity to the Number of Bins in the Chow-Liu Constraint

We investigate the sensitivity of the Chow-Liu constraint to the number of discretization bins $B$ on the 12 datasets used in our main experiments. The following plots summarize empirical coverage, fidelity, pruning rate, and runtime as functions of the miscoverage level $\alpha$. Across $B \in \{4, 8, 16\}$, the qualitative behavior remains stable: coverage tracks the conformal target, fidelity remains high, and the main trade-off is between a finer distributional representation and a heavier Oracle. We use $B = 4$ in the main experiments because it gives a compact MILP encoding while preserving the pruning-fidelity pattern observed with larger bin counts.

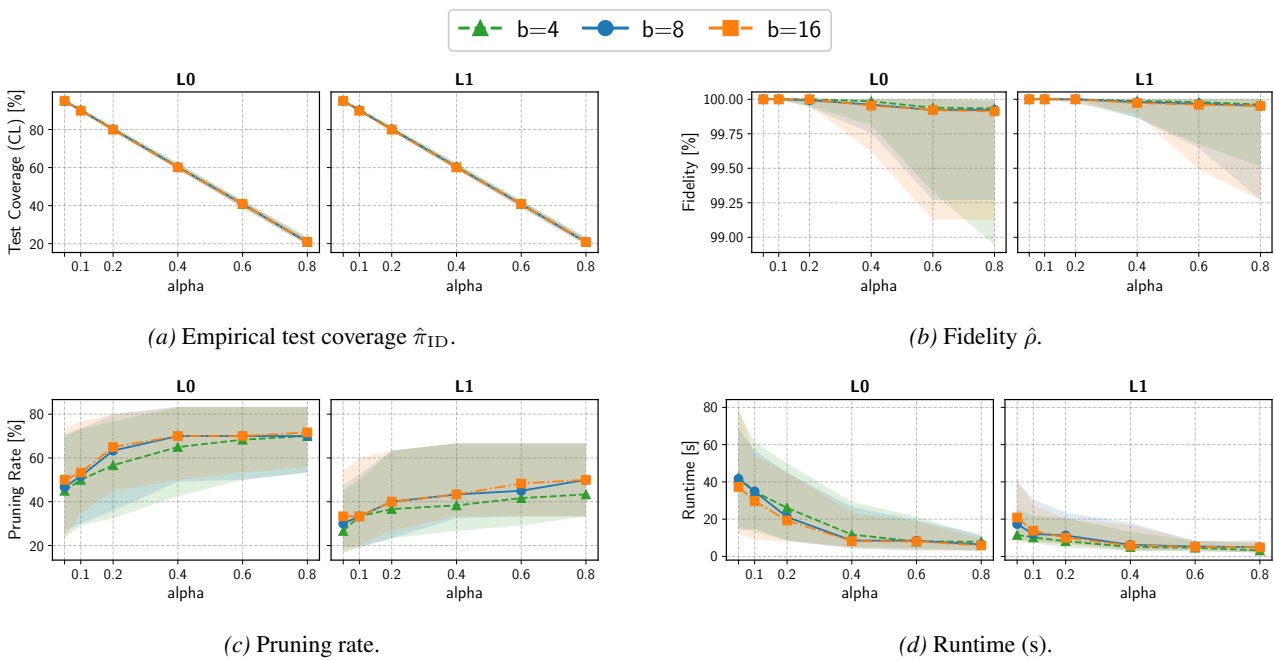

*(a)* Empirical test coverage $\hat{\pi}_{\mathrm{ID}}$.

*(b)* Fidelity $\hat{\rho}$.

*(c)* Pruning rate.

*(d)* Runtime (s).

*Figure 12.* Sensitivity of PINE-CL to the number of discretization bins $B$ used in the Chow-Liu constraint (12 datasets).

## B.7. Additional Results: XGBoost Larger Ensembles ($M = 50$) and Random Forests ($M = 30$)

We additionally report results on larger XGBoost ensembles ($M = 50$) under the $L_0$ objective.

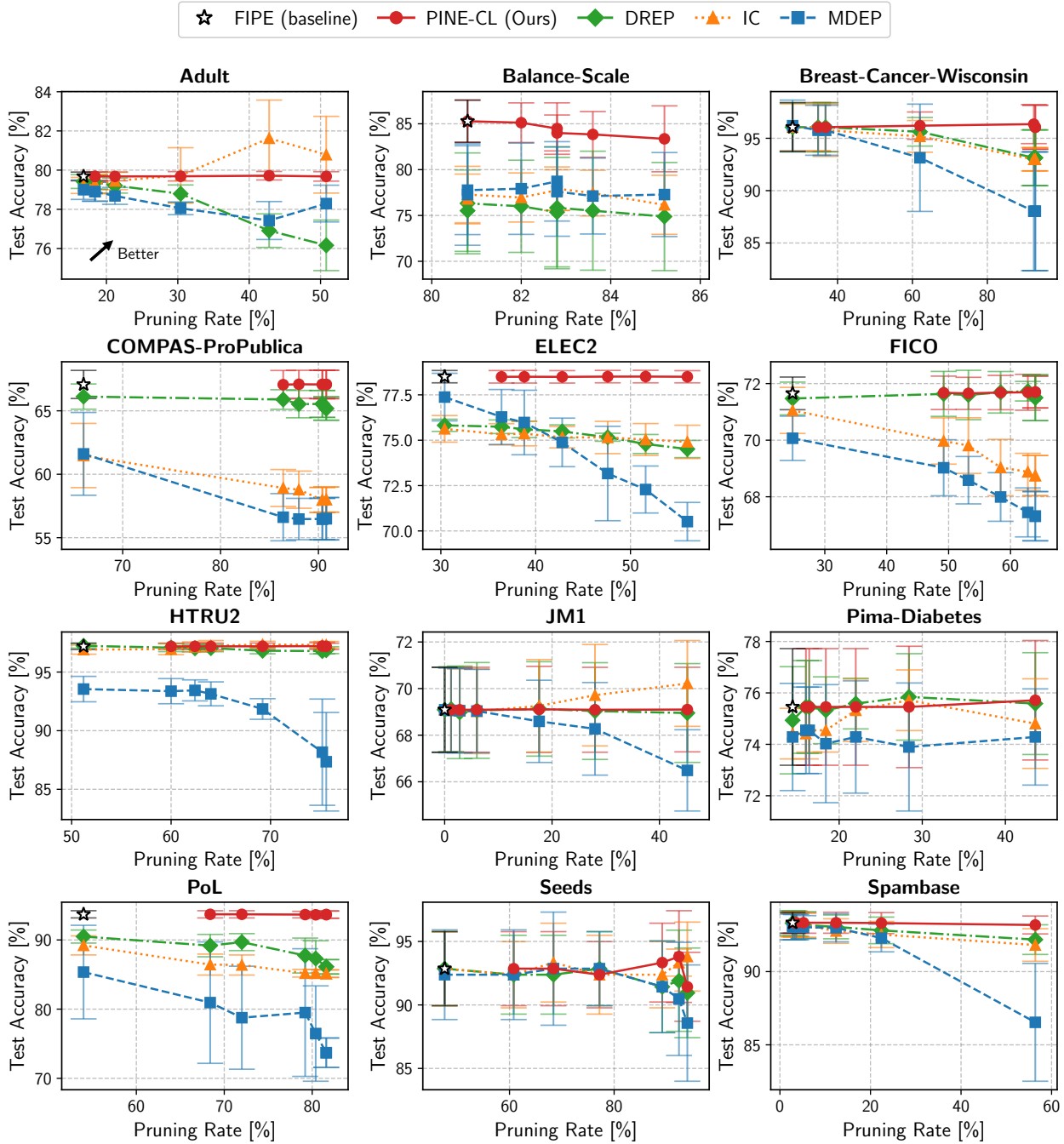

*Figure 13.* (XGBoost, $M = 50$, $L_0$) Test Accuracy - Pruning Rate (12 datasets).

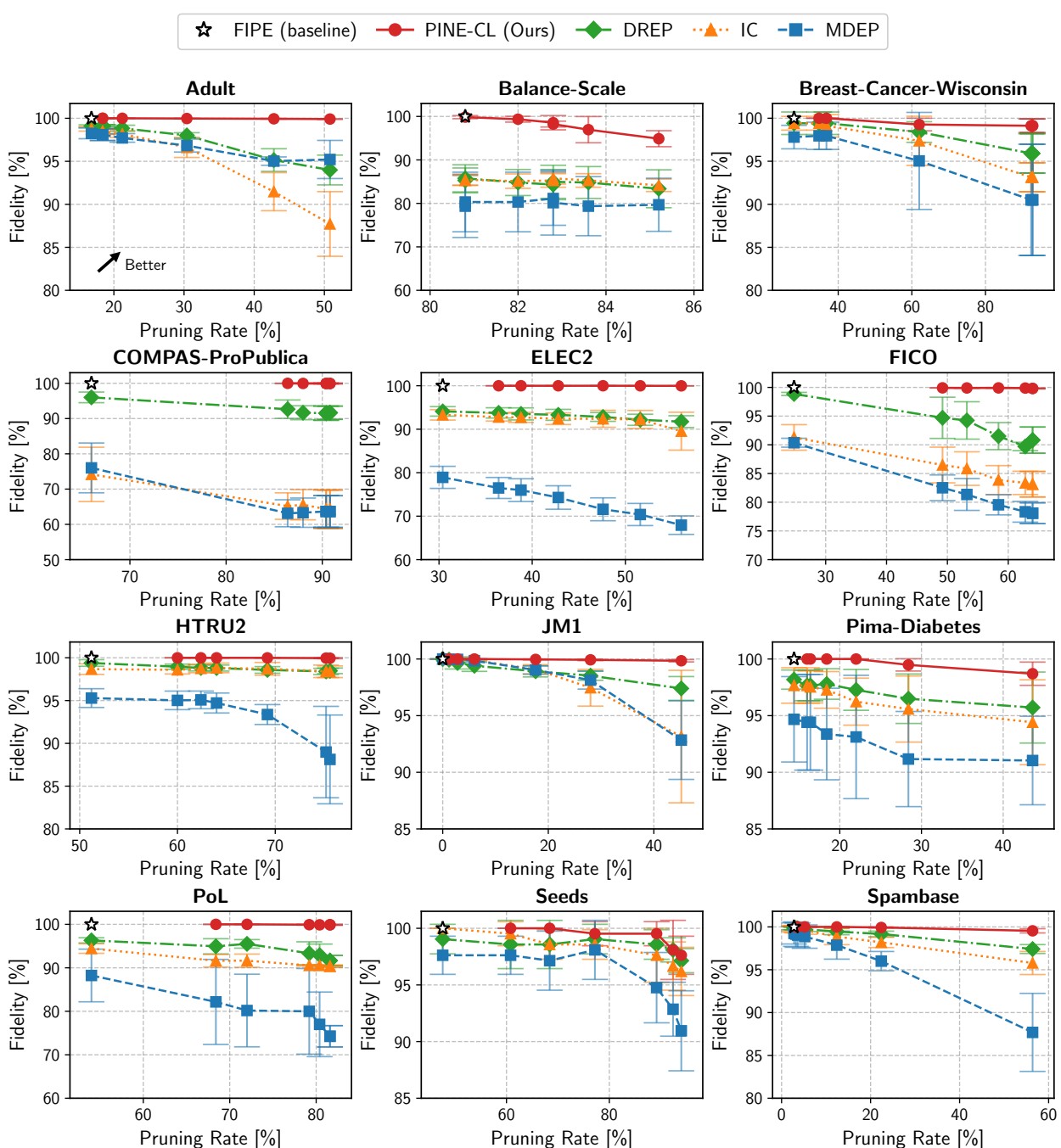

*Figure 14.* (XGBoost, $M = 50$, $L_0$) Fidelity - Pruning Rate (12 datasets).

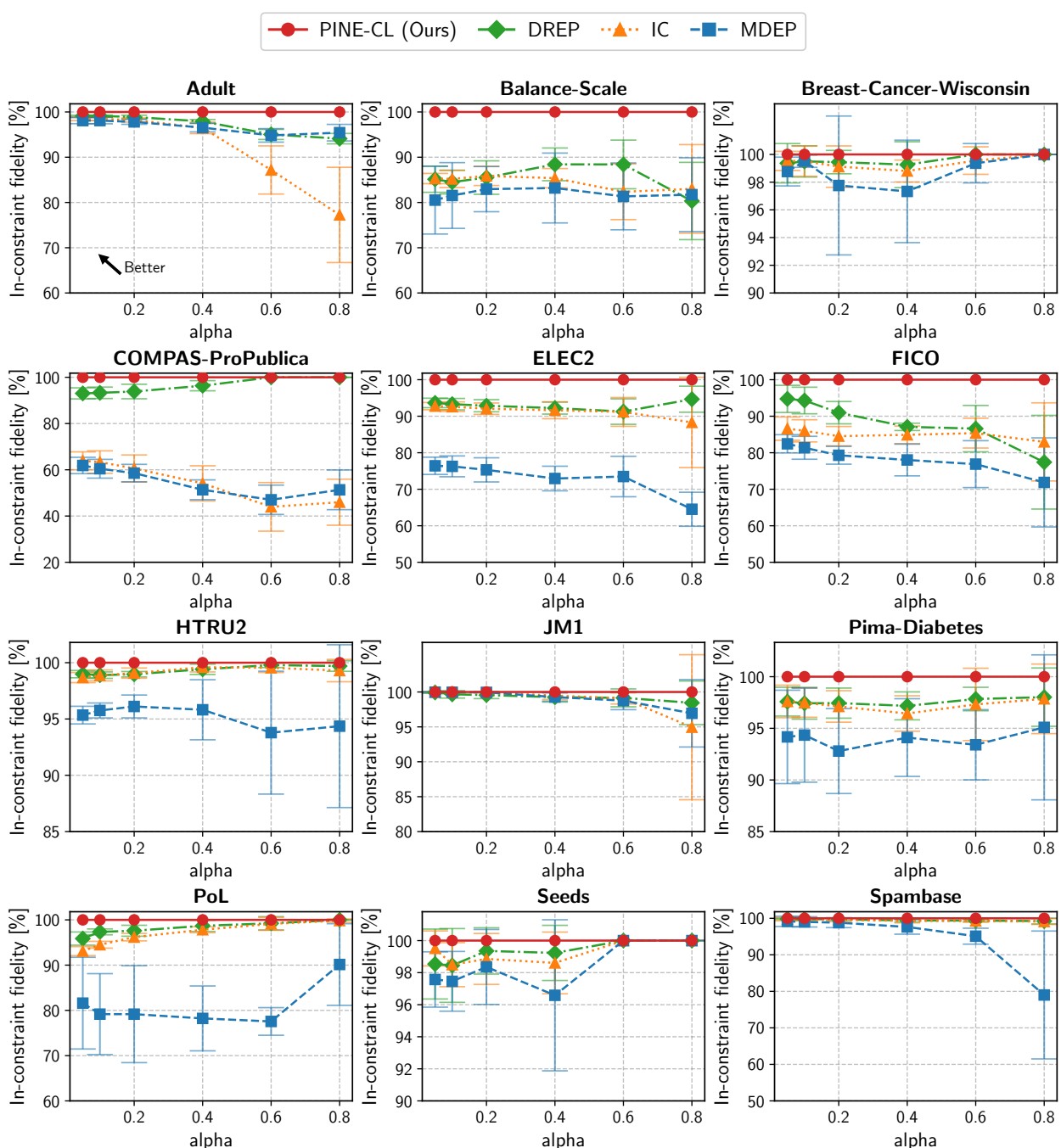

*Figure 15.* (XGBoost, $M = 50$, $L_0$) In-constraint Fidelity - alpha (12 datasets).

We additionally report results on Random Forests ($M = 30$) under the $L_0$ objective.

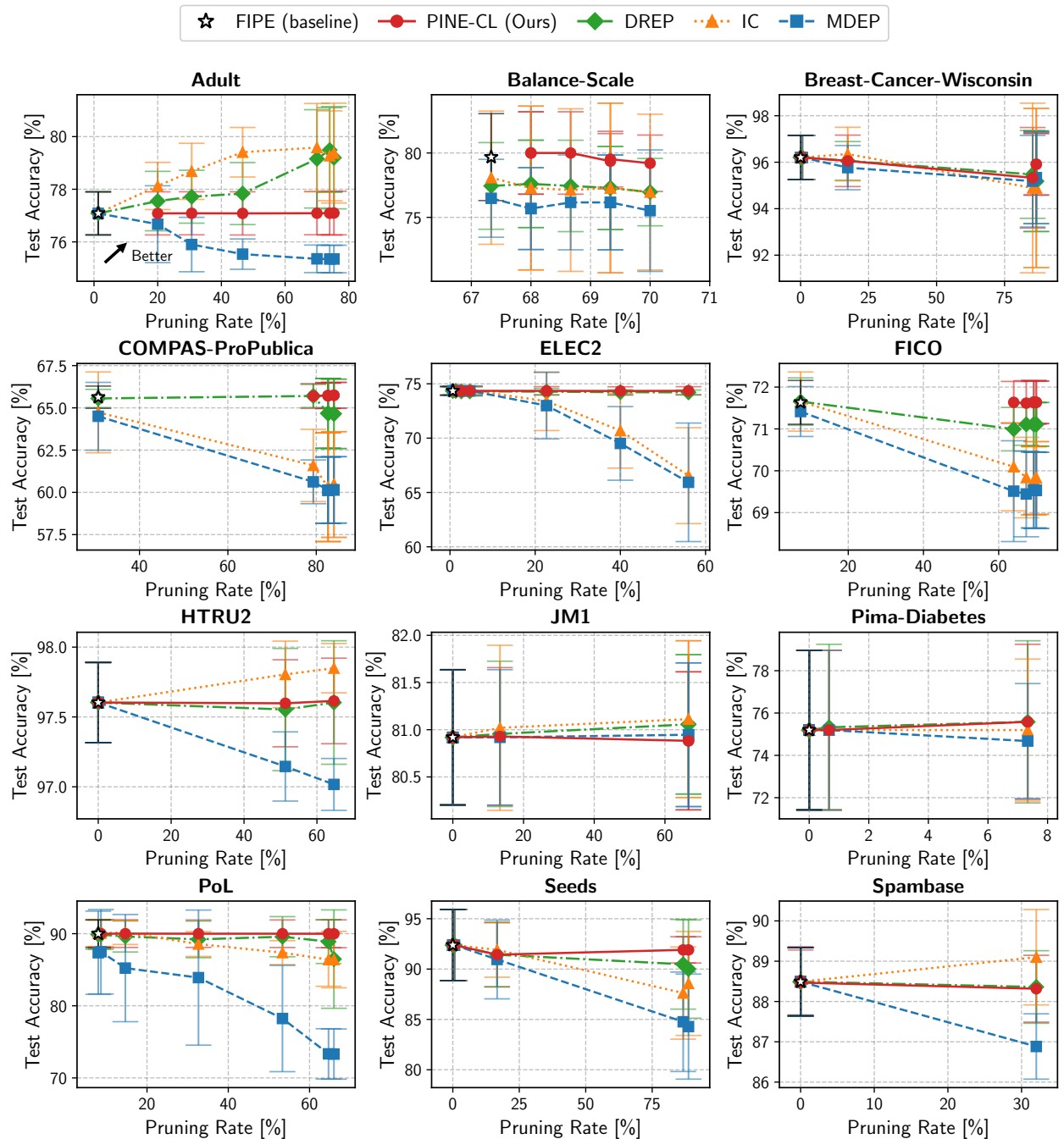

*Figure 16.* (Random Forest, $M = 30$, $L_0$) Test Accuracy - Pruning Rate (12 datasets).

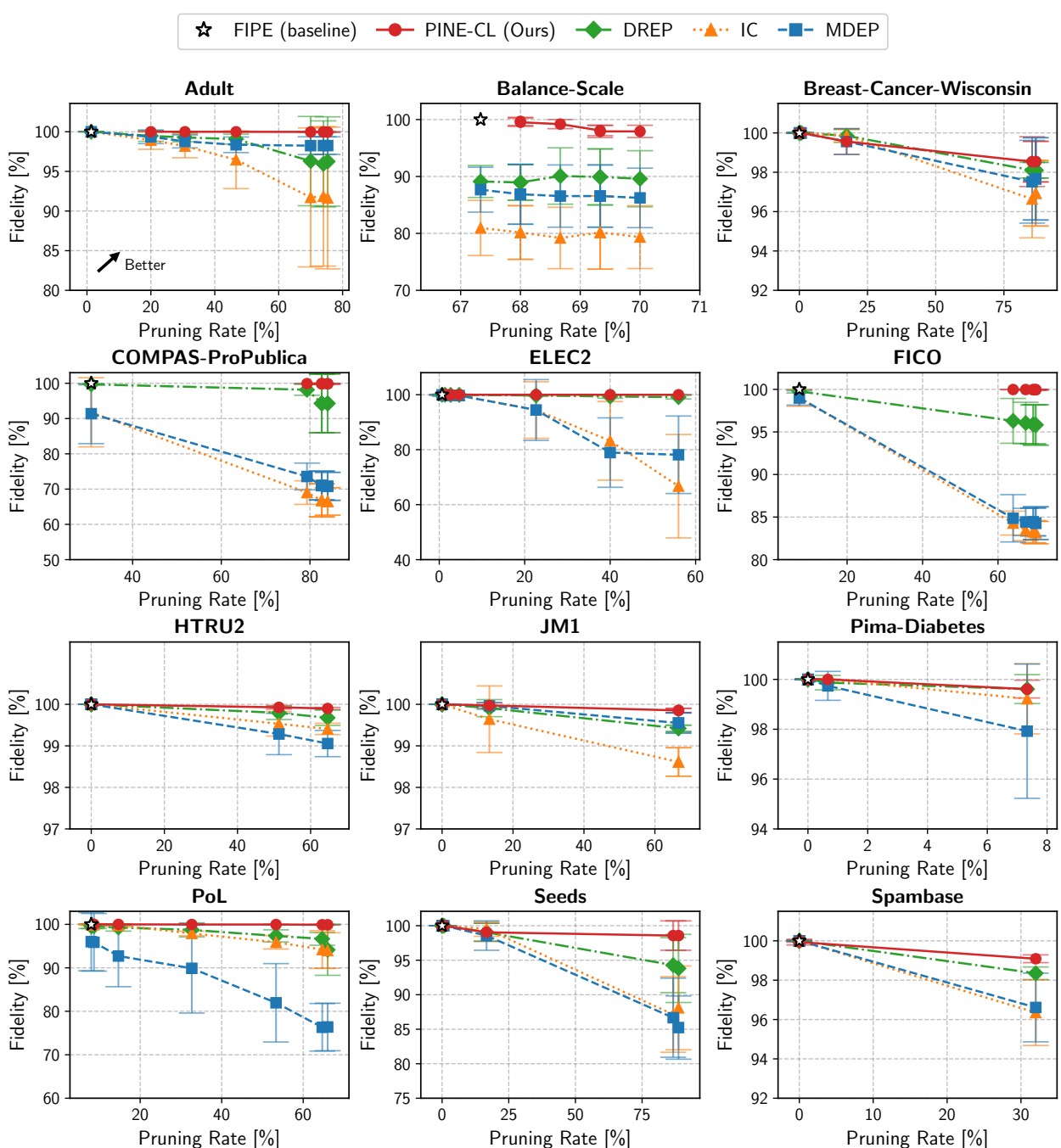

*Figure 17.* (Random Forest, $M = 30$, $L_0$) Fidelity - Pruning Rate (12 datasets).

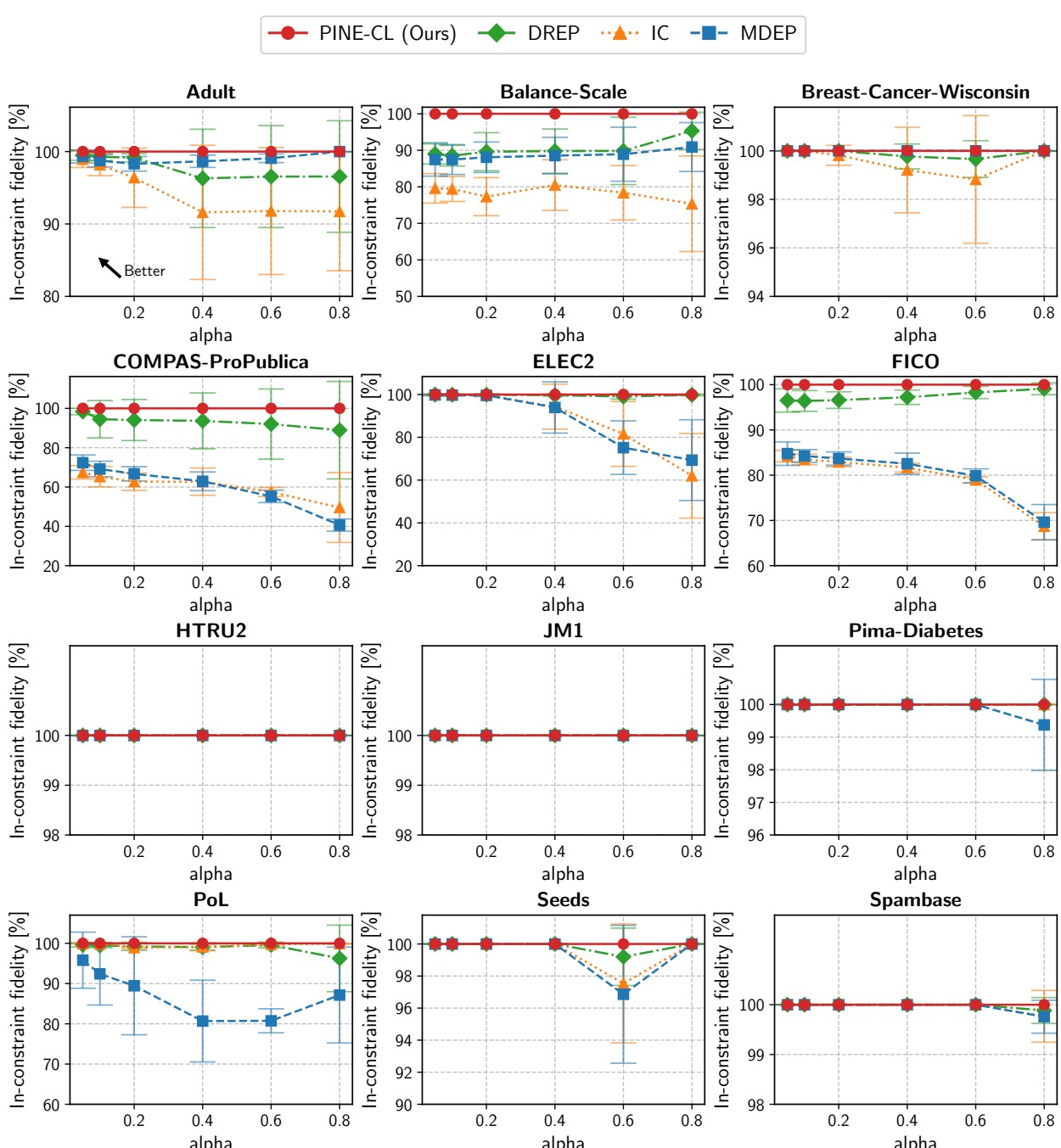

*Figure 18.* (Random Forest, $M = 30$, $L_0$) In-constraint Fidelity - alpha (12 datasets).

## C. Plausible Scores Representable as MILP Constraints

To restrict the Oracle search region to the in-distribution region $\mathcal{X}_{\mathrm{ID}}(\alpha) = \{\boldsymbol{x} \mid s(\boldsymbol{x}) \leq \tau(\alpha)\}$, we must be able to express the constraint $s(\boldsymbol{x}) \leq \tau(\alpha)$ as linear constraints in a MILP. Accordingly, PINE uses scores that (i) can be estimated from $\mathcal{D}_{\mathrm{fit}}$ without labels and (ii) are based on tree structure (or tree-representable models), making them amenable to MILP encoding. Below we summarize the definition and MILP embedding of Leaf Support (LS) and Isolation Forest (IF). (Chow-Liu tree (CL) is described in Section 4.1; implementation details for rounding bin boundaries are provided below.)

**Leaf Support (LS).** Leaf Support measures in-distribution plausibility via leaf visitation frequency, based on the idea that inputs that reach frequently visited leaves on the training data are more likely under the data distribution. For a leaf $\ell \in \mathcal{L}_m$ of tree $T_m$, we define the empirical leaf visitation count on $\mathcal{D}_{\mathrm{fit}}$ and its (smoothed) ratio as

$$
\begin{aligned}
n_{m,\ell} &:= |\{\boldsymbol{x} \in \mathcal{D}_{\mathrm{fit}} \mid \lambda_m(\boldsymbol{x}) = \ell\}|, \\
p_{m,\ell} &:= \frac{n_{m,\ell} + \beta}{\sum_{\ell' \in \mathcal{L}_m} n_{m,\ell'} + \beta |\mathcal{L}_m|},
\end{aligned}
\tag{18}
$$

where $\beta > 0$ is a smoothing coefficient that ensures $p_{m,\ell} > 0$ even for leaves that are never reached in $\mathcal{D}_{\mathrm{fit}}$. Let $a_{m,\ell} := -\log p_{m,\ell}$. Then the score of input $\boldsymbol{x}$ is given as the sum of the $a_{m,\ell}$ values corresponding to the leaves reached by $\boldsymbol{x}$ in each tree:

$$
s_{\mathrm{LS}}(\boldsymbol{x}) := \sum_{m=1}^{M} a_{m,\lambda_m(\boldsymbol{x})}.
\tag{19}
$$

In the MILP, we introduce a binary variable $z_{m,\ell} \in \{0,1\}$ indicating whether input $\boldsymbol{x}$ reaches leaf $\ell$ in tree $m$ ($z_{m,\ell} = 1 \Leftrightarrow \boldsymbol{x} \in R_{m,\ell}$). Using these leaf-indicator variables, Equation (19) can be written as a linear sum over $z_{m,\ell}$, so the in-distribution constraint $s_{\mathrm{LS}}(\boldsymbol{x}) \leq \tau(\alpha)$ can be added directly as a single linear inequality.

**Isolation Forest (IF) (Liu et al., 2008).** Isolation Forest is an anomaly detection method that generates many isolation trees by repeatedly splitting data using random features and split points, and measures how easily a point can be isolated (i.e., how anomalous it is) by the path length from the root to the leaf corresponding to the point. In general, out-of-distribution points tend to be separated with fewer splits and thus have shorter path lengths, whereas in-distribution points are harder to isolate and have longer path lengths. PINE uses this property and regards points with sufficiently long path lengths as in-distribution.

Concretely, we train $K$ isolation trees on $\mathcal{D}_{\mathrm{fit}}$ and assign to each leaf $\ell \in \mathcal{L}_k$ of tree $I_k$ the (corrected) path length $h_{k,\ell}$ attained when $\boldsymbol{x}$ reaches that leaf. In Isolation Forest, if splitting stops while multiple points remain in a leaf, path lengths can be underestimated; it is standard to add a correction based on the expected search length in a binary search tree:

$$
c(n) := 2H_{n-1} - \frac{2(n-1)}{n},
\tag{20}
$$

where $n$ denotes the number of training points remaining in the leaf and $H_k := \sum_{i=1}^{k} 1/i$ is the $k$-th harmonic number. In PINE, we precompute $h_{k,\ell} := \mathrm{depth}_k(\ell) + c(n_{k,\ell})$, where $n_{k,\ell}$ is the number of points in $\mathcal{D}_{\mathrm{fit}}$ that fall into leaf $\ell$ of $I_k$, and store it as a per-leaf constant.

In the MILP, we introduce a binary variable $g_{k,\ell} \in \{0,1\}$ indicating whether $\boldsymbol{x}$ reaches leaf $\ell$ in tree $I_k$, and enforce $\sum_{\ell \in \mathcal{L}_k} g_{k,\ell} = 1$. Then $\sum_{\ell \in \mathcal{L}_k} h_{k,\ell} g_{k,\ell}$ represents the corrected path length in tree $I_k$ for input $\boldsymbol{x}$. We can thus define the average path length as

$$
L(\boldsymbol{x}) := \frac{1}{K} \sum_{k=1}^{K} \sum_{\ell \in \mathcal{L}_k} h_{k,\ell}\, g_{k,\ell}.
\tag{21}
$$

To match the convention that smaller scores correspond to more in-distribution points, we set $s_{\mathrm{IF}}(\boldsymbol{x}) := -L(\boldsymbol{x})$. The in-distribution constraint can then be added to the MILP as $s_{\mathrm{IF}}(\boldsymbol{x}) \leq \tau(\alpha)$. Note, however, that IF introduces additional binary variables and constraints proportional to the number of trees $K$, which can make the Oracle MILP heavier.

**Rounding bin boundaries for the Chow-Liu tree.** In the Chow-Liu tree (CL) described in Section 4.1, we discretize continuous features into $B$ bins and then learn the CL model. Since bin boundaries must be representable in the Oracle

MILP (i.e., consistent with existing split-indicator variables), we round each boundary to the set of split thresholds $\Theta_j$ used by the trained ensemble.

Concretely, for a continuous feature $j$, we first compute the target bin boundaries for a $B$-way split using empirical quantiles on $\mathcal{D}_{\text{fit}}$:

$$\kappa_{j,b} := \text{Quantile}_{b/B}\big(\{x_j \mid \boldsymbol{x} \in \mathcal{D}_{\text{fit}}\}\big),\ b = 1, \ldots, B - 1. \tag{22}$$

Next, given the sorted list of thresholds in $\Theta_j$, we map each $\kappa_{j,b}$ to the nearest split value as follows:

$$\hat{\kappa}_{j,b} := \underset{t \in \Theta_j}{\arg\min} |t - \kappa_{j,b}|. \tag{23}$$

In the case of ties, we choose the larger split value. Finally, we remove duplicates and sort the boundary list $\{-\infty, \hat{\kappa}_{j,1}, \ldots, \hat{\kappa}_{j,B-1}, +\infty\}$. If rounding introduces duplicate boundaries, the effective number of bins can be smaller than $B$; in our implementation, we learn the CL model with the reduced state count (and exclude features for which only $-\infty$ and $+\infty$ remain).

In terms of computational complexity, extracting and sorting $\Theta_j$ takes $\mathcal{O}(|\Theta_j| \log |\Theta_j|)$, and rounding each boundary can be done via binary search in $\mathcal{O}(\log |\Theta_j|)$. Thus, for each feature $j$ the procedure runs in $\mathcal{O}(|\Theta_j| \log |\Theta_j| + (B - 1) \log |\Theta_j|)$.

# D. Comparison with Other Plausible Scores

All results in this section use XGBoost ensembles with $M = 30$ trees and compare plausible scores: Chow-Liu (CL), Leaf Support (LS), and Isolation Forest (IF). The score-comparison experiments show that PINE is not tied to a single plausibility score, but that score choice changes the trade-off among runtime, coverage, and pruning. Chow-Liu is used as the main instantiation because it captures pairwise feature dependence while remaining compact in the MILP; Leaf Support and Isolation Forest provide simpler or more flexible alternatives, but they can change the size and difficulty of the Oracle problem.

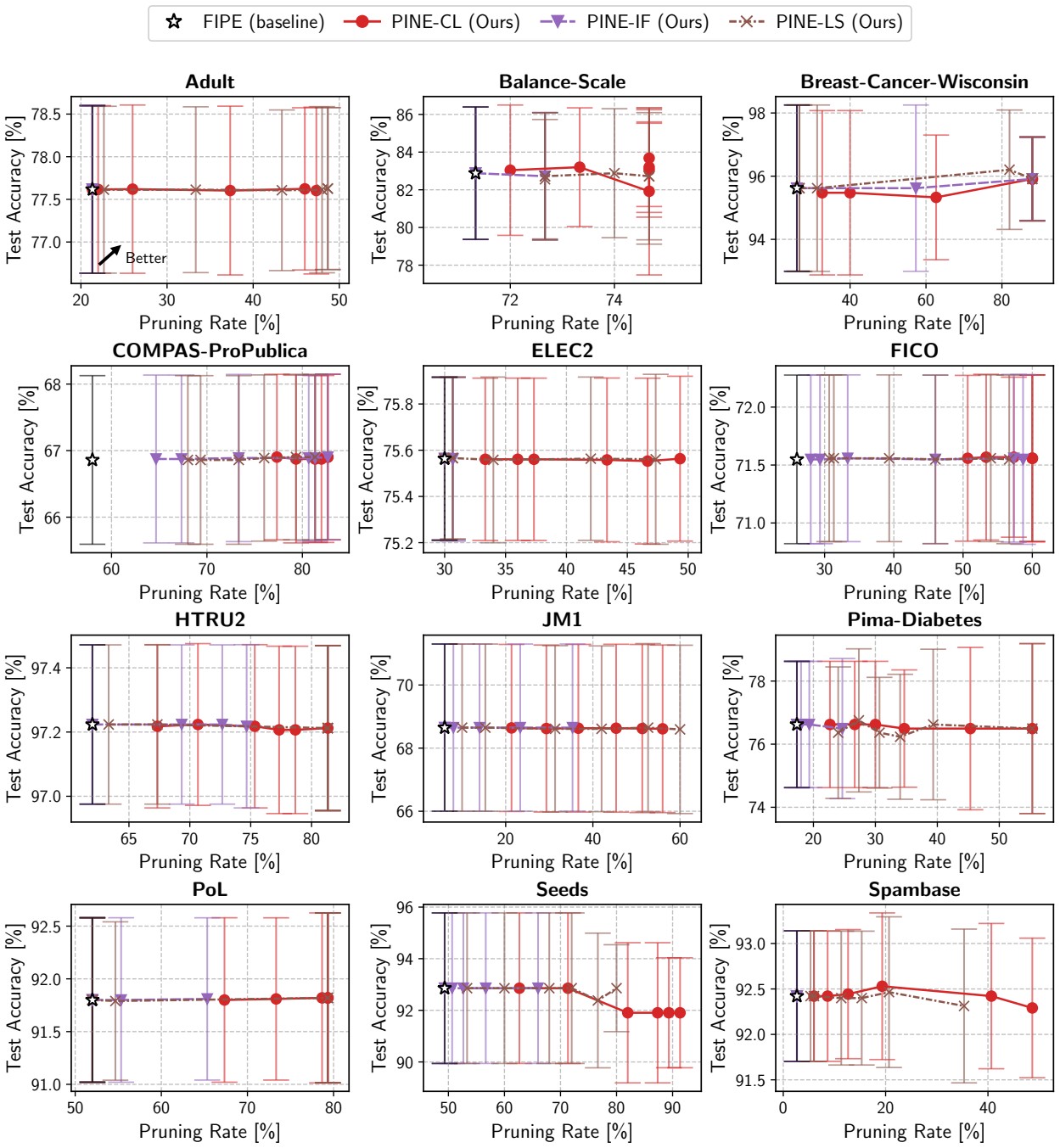

*Figure 19.* (XGBoost, $M = 30$, $L_0$) Test Accuracy - Pruning Rate (12 datasets).

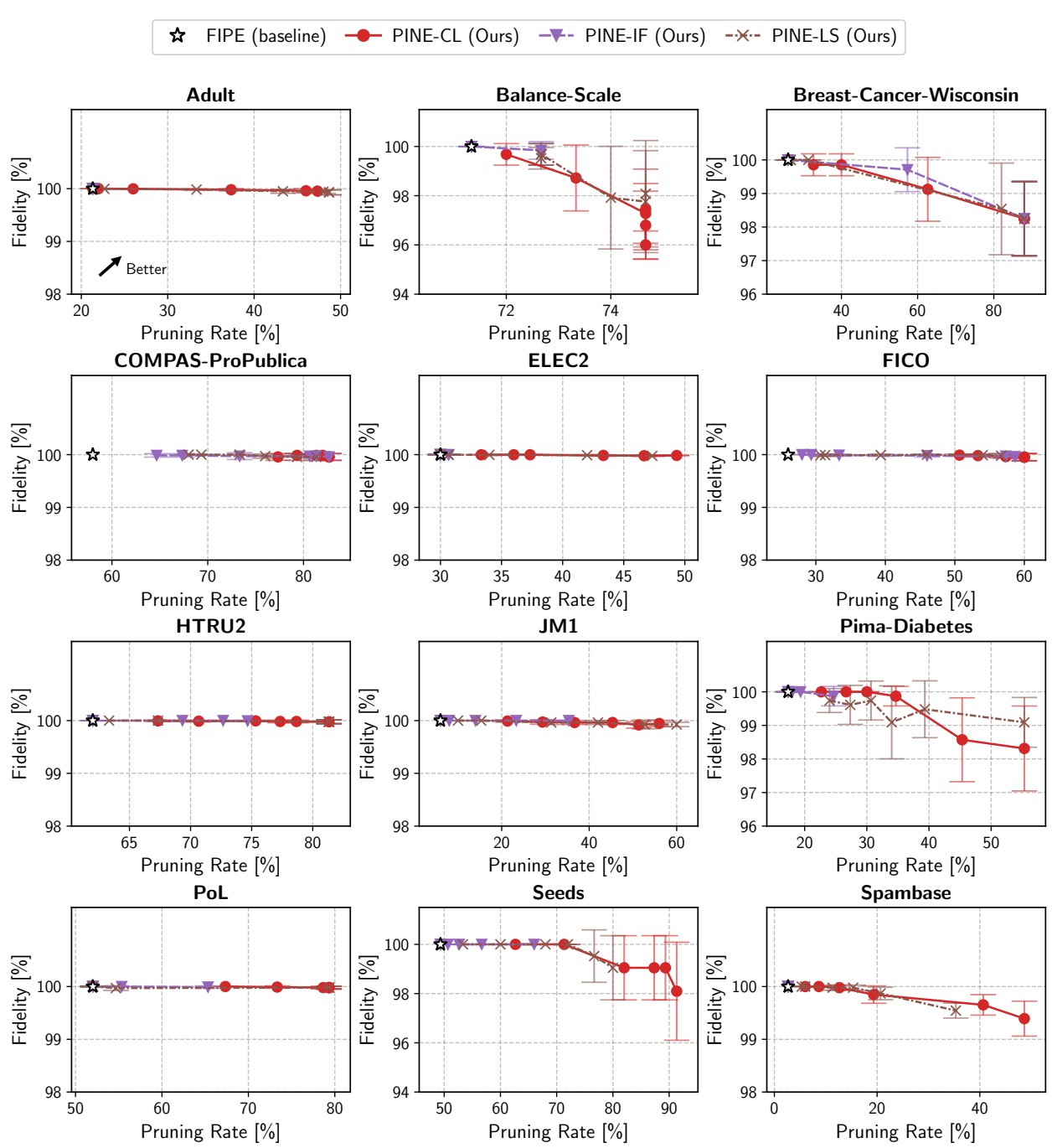

*Figure 20.* (XGBoost, $M = 30$, $L_0$) Fidelity - Pruning Rate (12 datasets).

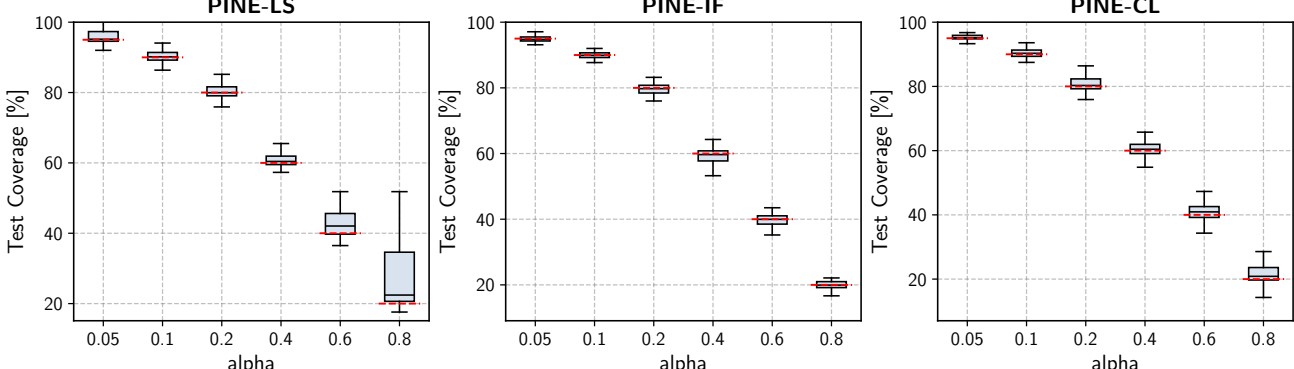

*Figure 21.* (XGBoost, $M = 30$, $L_0$) Coverage vs. miscoverage level alpha (12 datasets).

