# OpenReview forum: "PINE: Pruning Boosted Tree Ensembles with Conformal In-Distribution Prediction Equivalence"
_ICML.cc/2026/Conference — ICML 2026 regular_

### Official Review · Reviewer_GhSa · 2026-02-19

**Soundness:** 4
**Presentation:** 3
**Significance:** 2
**Originality:** 2
**Overall Recommendation:** 4
**Confidence:** 5

**Summary:**

This paper presents an extension of FIPE framework for fidelity based pruning of tree ensembles. They do this by using a tree-based probability model based on Chow-Liu trees for a plausibility score and split-based conformal prediction to decide in vs. out of distribution.

**Compliance With Llm Reviewing Policy:**

Affirmed.

**Final Justification:**

For me, the paper is quite borderline. The reasons to accept are:

* The paper is well written and clear. The authors are very thoughtful, particularly in their rebuttals. They bring good (extra) argumentation

* After the rebuttal, the experimental insights are quite good and interesting.

* This is an interesting / relevant area

The reasons not to are

* There are quite a few modifications to make. The justification in the intro needs to be reworked. The paper will have to changed quite a bit even with 1 extra page to get all the relevant details into it, particularly with respect to the algorithmic background (less important imo but other reviewers want this) and experimentations. This is particularly true given that main body gives a very narrow and cursory overview of the results.  This are quite big changes (e.g., versus just adding one extra baseline) that should be done.

* To me, the contribution feels on the narrower side (also wrt applicability given some of the limitations / experimental results)

Ultimately, I lean towards acceptance because the (a) the authors are sufficiently thoughtful and honest in the rebuttal that I think they'll do the reworking (which is really necessary), (b) there are some interesting empirical insights, and (c) I think it is good to maintain some diversity wrt to topics.

**Key Questions For Authors:**

I have a couple of questions about the statement “We exclude node-level pruning…because they alter the tree structure and would require re-encoding constraints for our Oracle-based equivalence evaluation.” Namely, I’d like to raise the following points:

*I presume that this has to with pruning the ensemble & not evaluating a pruned ensembles performance (fidelity / accuracy) on unseen data. I.e., the authors want to use their fidelity based criteria to guide the pruning process as opposed to the process proposed in ForestPrune and LOP. I am correct?

*Is this extension infeasible? Difficult?

A key question I would have is about the scalability of the approach. Namely, the depth of the trees is limited to 2 which is quite shallow as most applications would require deeper trees or including more trees in the ensemble.
The datasets tend to be smaller both wrt features and examples. I.e., would expect simpler models to work well here.

**Limitations:**

Yes.

**Strengths And Weaknesses:**

I appreciate the simplicity of the idea and the extremely clear writing of the paper. The authors set up the work in an intuitive and easy to follow manner. The decision choices are sensical and well motivated.

The work does come across as an extension of FIPE so it it is a smaller step conceptually in that regard.

I am not sure that the authors have best used the space available in the main body of the paper. The results presented in the paper are quite limited focusing on just 1 of the considered datasets.

In terms of freeing up space, I think Figure 1 could convey the same amount of information but use less pass (e.g., for me the FIPE part could go and you could then really get away with one image).  Figure 2 also probably has some extraneous information; note here that the color shading isn’t explained.

---after rebuttal---

The extra experimentation is very interesting. The approach works quite for ensembles with restricted tree depth; it is quite interesting that it does better than forestprune in this setting. I'm not surprised that for deeper ensembles forest prune does well. It also makes sense that pine doesn't prune much for deeper ensembles as the more fine-grained differentiation offered by deeper trees probably means you need to lose fidelity to prune.

I think the motivation for the importance of fidelity could be sharper. This is clearly the key criteria if you are deploying the big ensemble but analyzing the pruned one for e.g., providing explanations, verifying it, etc.

If the goal is reducing memory use or inference time, maintaining fidelity strikes me as being less important. Here fidelity is a more restrictive version of comparing the accuracies of the original and pruned ensemble. So if you just want smaller models then the node-based approaches will give much better compression.

I also see some risks for looking at in distribution. In many high-risk applications, the data can be quite biased. I think many medical datasets are not representative of the populations at large (e.g., it is harder to get permission for data about children but models trained only using data from adults can be applied to children).

---

> ### Author Rebuttal · Authors · 2026-03-29
>
> Thank you for the clear review. The comments on structure, evaluation, and motivation are well taken, and we will reflect them in the final manuscript. We focus on two key questions below.
>
> **1. About excluding node-level pruning / ForestPrune / LOP.**
>
> The reviewer's interpretation is largely correct, and the extension is difficult but not infeasible. The issue is not post-hoc evaluation of a fixed compressed model: a frozen ForestPrune/LOP model could still be evaluated after compression for test accuracy, fidelity, or conditional fidelity on X_ID(α). What is non-trivial is using prediction equivalence to guide the pruning process itself.
>
> In the current PINE/FIPE formulation, tree topology and leaf score vectors are fixed, the optimization variable is only the nonnegative tree-weight vector w, and the Oracle searches for counterexamples for the current w via a MILP over leaf/path selections. So the extension is not infeasible in principle, but it is not direct under the current formulation. ForestPrune is the more natural next step because it introduces per-tree contiguous depth-cut decisions. LOP is more involved: it can prune arbitrary subtrees and simultaneously update leaf values, so both active structure and prediction scores change during optimization. Incorporating such operators into a pruning loop guided by prediction equivalence would therefore require a new Oracle/Pruner that jointly encodes structural decisions, path consistency, and resulting prediction values. Among these, depth-wise truncation as in ForestPrune appears to be the more direct extension. We will clarify in the paper that the issue is compatibility with the current Oracle, not whether these methods can be evaluated after compression.
>
> **2. About scalability, shallow trees, and whether simple models may already suffice.**
>
> PINE remains usable on the larger/deeper ensembles we tested, but runtime grows mainly with the difficulty of Oracle counterexample search, so the number of trees and depth matter more than dataset size alone. The shallow/small main-paper setting was a benchmark choice for a fully certified multi-dataset comparison, not a framework limitation. The reason is simple: larger/deeper ensembles enlarge the Oracle counterexample-search problem, which tends to increase the number of Oracle calls and solve time. You can see this first in the formulation and then in the additional experiments.
>
> At the formulation level, the Oracle MILP per ordered class pair has size O(M*2^D + pB^2): M and D drive the ensemble-side search size, while p and B govern the Chow-Liu encoding size. Alpha is a separate difficulty knob because it changes how large a region must be certified, but the main structural drivers here are still M and D. So the practical issue is structural complexity, not dataset size alone, and in this slice training-set size is secondary while the feature dimension p is minor.
>
> Our additional experiments support exactly this picture. At fixed α=0.8, mean fidelity stays in 99.01%-99.29% across the completed larger/deeper settings, and even (M,D)=(50,3), where M and D denote the number of trees and max depth, still reaches 47.0% pruning with 99.28% fidelity. Runtime grows from (M,D)=(30,2)->4.29s to (50,2)->165.11s, (70,2)->921.38s, and (100,2)->3280.48s; at depth 3 it is 44.03s, 1206.33s, and 13148.44s for M=30,50,100. Depth therefore matters sharply at matched M: depth 3 is 10.3x, 7.3x, and 4.0x slower than depth 2 at M=30,50,100. These are additional larger/deeper settings, not just larger dataset sizes. The runtime-driver analysis is consistent: the number of iterations has Spearman ρ=+0.909; among user-facing factors, M*2^D has ρ=+0.508, M has ρ=+0.479, training-set size has ρ=+0.343, depth D has ρ=+0.132, and input dimension p has ρ=+0.083. These are descriptive correlations rather than a full causal decomposition, but they align with the formulation-level picture above. Among user-facing structural quantities, the number of trees is therefore the strongest direct runtime driver in this slice, while the number of iterations is the main downstream marker of search difficulty. This also explains the conservative main-paper design: we prioritized a fully certified multi-dataset comparison at a common scale, then evaluated additional larger/deeper settings separately.
>
> We also agree that if the goal were only memory or latency, simpler or node-based methods could suffice. But Fig. 3 shows that similar test accuracy can coexist with substantially different fidelity, so that is a different target from compressing a fixed deployed ensemble while keeping its predictions unchanged. On this target, PINE remains stronger in aggregate fidelity against both simple heuristics (reviewer BebJ, Q2/5) and all 15 documented PyPruning baselines (reviewer JqYT, Q3), suggesting that its advantage is not just an artifact of the smallest setting. We will revise the paper to make this distinction clearer.

---

> > ### Author Rebuttal · Reviewer_GhSa · 2026-04-02
> >
> > The authors bring extra, thoughtful, well articulated and reasoned arguments. I very much appreciate this.
> > Conceptually, the leap still remains smaller (as also mentioned by Bebj).
> >
> > For the scaling, I find increasing the number of trees far less important. In my experiences on practical problems, there's rarely a reason to go beyond 100, with 50 often being quite good. (I think this shows up in benchmarks too if one looks closely at learning curves.). Thus the depth seems to be the bigger issue and indeed the complexity argument hints that this may be a problem as does the experiments look at depth 2 and 3.
> >
> > If the criticism as mentioned by JqYT is that a broader set of baselines are needed in the experiments, then seeing e.g., forestprune (using  an accuracy bound in place of a faithfulness bound) and check its faithfulness post hoc. These approaches will likely give much more aggressive pruning than the baselines in PyPrune.

---

> > > ### Author Response · Authors · 2026-04-06
> > >
> > > We thank the reviewer very much for the thoughtful follow-up comments. We address the remaining points most relevant to the reviewer's concerns.
> > >
> > > **(i) Scalability.**
> > >
> > > To test depth-related scalability, we conducted additional PINE experiments with α=0.8 over (M,D) in {10,20,30,40,50} x {2,3,4,5}, where M is the number of trees and D is the maximum depth. The results show that depth is the main bottleneck for attainable pruning rate, while larger M mainly increases runtime and has a much smaller effect on pruning rate. We summarize the mean behavior over all 12 datasets below.
> > >
> > > |Depth|Mean tree pruning rate [%]|Mean fidelity [%]|Mean in-constraint fidelity [%]|Mean runtime [s]|Mean number of iterations|Settings with almost no pruning (<=1% tree pruning)|
> > > |---|---:|---:|---:|---:|---:|---:|
> > > |D=2|64.78|99.39|100.00|51.86|3.23|0/60|
> > > |D=3|47.12|99.44|100.00|229.02|7.12|3/60|
> > > |D=4|36.57|99.06|100.00|975.99|12.07|19/60|
> > > |D=5|31.93|99.21|100.00|2162.50|13.67|29/60|
> > >
> > > At fixed M, mean pruning rate decreases by 29.58 to 35.83 pp from D=2 to D=5, while mean fidelity changes by only -0.63 to +0.46 pp. Over the same change in depth, runtime grows by x22.55-x603.78 and the number of iterations grows by x3.30-x6.08. The increase in settings with almost no pruning is also consistent with PINE's tree-level pruning objective: as trees become deeper, more trees remain partially useful and are harder to remove entirely. The core pattern is therefore not a fidelity collapse, but a compression collapse under much heavier runtime and optimization effort.
> > >
> > > By contrast, at fixed depth, increasing M from 10 to 50 changes mean pruning rate by only +7.83 to +10.00 pp. This is why we view depth as the sharper practical bottleneck for compression: both M and D increase runtime, but depth is what most directly erodes attainable pruning.
> > >
> > > **(ii) Stronger baseline.**
> > >
> > > To broaden the baseline comparison, we conducted additional post-hoc experiments with ForestPrune on 10 binary-classification datasets. The main trade-off is clear: ForestPrune is stronger for size/speed, while PINE is stronger for post-hoc faithfulness to the original ensemble. In our experiment, ForestPrune used the same data splits as PINE. Here, abserr denotes the allowed accuracy drop on the validation set relative to the original ensemble, whereas the accuracy change below is measured on the test set; the table reports the conservative ForestPrune point with abserr=0.005. Leaf pruning rate denotes the fraction of removed leaves.
> > >
> > > |Setting|Method|Leaf pruning rate [%]|Overall fidelity [%]|In-constraint fidelity [%]|Accuracy change [pp]|Runtime [s]|
> > > |---|---|---:|---:|---:|---:|---:|
> > > |(M,D)=(30,2)|PINE (α=0.8)|63.64|99.42|100.00|+0.06|3.91|
> > > |(M,D)=(30,2)|ForestPrune (abserr=0.005)|58.05|96.65|99.24|-0.85|8.09|
> > > |(M,D)=(30,4)|PINE (α=0.8)|29.23|99.92|100.00|-0.01|200.34|
> > > |(M,D)=(30,4)|ForestPrune (abserr=0.005)|74.22|95.56|98.11|-1.32|12.07|
> > >
> > > At D=2, PINE is better on both fidelity and compression.
> > > At D=4, ForestPrune compresses much more aggressively and runs much faster, but loses noticeably on fidelity and accuracy.
> > > This is related to the same depth effect seen in (i): as trees become deeper, more trees remain partially useful and are harder for PINE to remove entirely, while ForestPrune can still prune within those trees at the leaf level.
> > > This is the main trade-off we want to make explicit in the revision: ForestPrune is preferable when the target is aggressive size/speed reduction under an accuracy bound, whereas PINE is preferable when the target is preserving the original ensemble's decisions on likely future inputs.
> > >
> > > **(iii) Contribution relative to FIPE.**
> > >
> > > To clarify PINE's contributions relative to FIPE, we summarize below what PINE adds beyond FIPE.
> > >
> > > **(a) Configurable plausibility-score-plus-threshold framework.** PINE separates the plausibility score s(x) from pruning and calibrates τ(α) by split conformal, so one α controls X_ID(α). The appendix gives Leaf Support and Isolation Forest as alternative scores.
> > >
> > > **(b) Chow-Liu as the main dependency-aware, MILP-compact instantiation.** Chow-Liu captures feature dependencies with additive univariate/bivariate terms while remaining compact in the MILP; by contrast, alternative scores such as Isolation Forest introduce extra binary variables and constraints and can make the Oracle heavier.
> > >
> > > **(c) New in-distribution guarantee relative to FIPE.** PINE guarantees exact prediction equivalence on X_ID(α). The theory makes two differences from FIPE explicit: because X_ID(α) ⊂ X, the optimal ||w||_0 is no worse than under FIPE; and under the Chow-Liu restriction, the Oracle search set satisfies |A_τ(α)| <= exp(τ(α)).
> > >
> > > In the revision, we will foreground these three points: the depth-scaling results, the stronger ForestPrune comparison, and PINE's contribution beyond FIPE. If these additions address the reviewer's remaining concerns, we would be grateful if the reviewer would consider updating the Overall Recommendation.

---

### Official Review · Reviewer_JqYT · 2026-03-06

**Soundness:** 3
**Presentation:** 3
**Significance:** 2
**Originality:** 2
**Overall Recommendation:** 5
**Confidence:** 5

**Summary:**

The paper introduces PINE, a fidelity pruning method with a stochastic bound. The method works as follows: A constraint set S is maintained (started with the training data) on which predictions between the pruned ensemble and the original one should be similar. Similarity is computed via a plausibility score $s$ and threshold $\tau$. $s$ is based on a Chow-Liu tree, whereas $\tau$ is indirectly set via a desired coverage $1-\alpha$ by the user (i.e. $\alpha$ is given by the user).
The algorithm is basically a combination of FIPE (i.e., iteratively prune trees via a pruner and reassign pruning samples based on $s$ via an oracle) and Chow-Liu tree construction plus the idea of coverage via $\alpha$. Based on the Chow-Liu tree construction and the coverage formulation, the author can show a bound on the number of oracle requests required to maintain a good in-distribution performance. Finally, the authors provide experiments on the performance that show how their method performs.

**Compliance With Llm Reviewing Policy:**

Affirmed.

**Final Justification:**

My two main concerns with the paper are

a) What is the exact setting of faithful pruning

b) Is the evaluation sound in the larger context of pruning

The rebuttal answered both questions adequately. If the paper gets accepted, I hope the authors have the space to include their new results in the Camera Ready and to discuss the setting a bit more in in-depth. Note: I updated my Overall Recommendation from "weak reject" to "accept," but I did not update any other score.

**Key Questions For Authors:**

1) What is your goal for pruning? You use only M=30 / M = 50 trees for the base ensembles. This is very small. For RF we typically see $\ge 256$ (see also https://link.springer.com/chapter/10.1007/978-3-642-31537-4_13) and for XGBoost I frequently come across $> 1000$ boosting rounds as well.
2) I do not understand the appeal behind faithful pruning (and by extension your theoretical analysis): Given eq 3 it is very simple to construct one tree that perfectly represents the decision boundary of the tree ensemble. This tree is very large, but it's just one tree. I understand that this is not in the spirit of pruning, but I am not sure what your goal is:
I always viewed pruning as a method to improve the ensemble accuracy while (hopefully) reducing its size (e.g. this is nicely captured by the "Many Could Be Better Than All"-theorem (https://www.sciencedirect.com/science/article/pii/S000437020200190X#:~:text=Ensembling%20neural%20networks:%20Many%20could%20be%20better%20than%20all%E2%98%86&text=This%20result%20is%20interesting%20because,as%20well%20as%20the%20variance.) Faithful pruning (and by extension your method) offers the same performance on in-distribution data, assuming we have enough training data / pruning data, as the original forest. Unless we have special constraints (time, energy, memory, interpretability by a human, etc) or we want to improve on out-of-distribution data, I don't see the benefit of why we prune. Could you please comment?
3) In your experimental analysis, it seems that you are better in roughly 50% of the cases (Note: This is by "eye"-balling, so please correct me here if I am wrong). This seems comparably weak for a new method that does not really have any other benefit (at-least none is shown). Moreover, you only compare it against a handful of methods. You mention PyPruning, which, according to its website, has 15 (?) implemented. Please comment on the overall performance of your method and the comparison against other methods.

**Limitations:**

Yes.

**Strengths And Weaknesses:**

Strength:
- Elegant novel pruning algorithm
- Well-written paper that is overall easy to follow

Weaknesses:
- Theoretical contribution that does not provide novel insights
- Experimental evaluation is weak
- No good argument why faithful pruning is more powerful than regular pruning (although that criticism also belongs to Emine etal. 2025)


More detailed review:

Overall, the paper is well-written and (mostly) easy to follow. I'd like Alg. 1 to be more self-contained, but this is more a matter of preference (i.e. what is the pruner and the oracle). The combination of Chow-Liu and FIPE is elegant, especially with the $\alpha$-coverage idea. The paper flow fits the ICML community and is of interest for the ensemble / ensemble pruning sub-community. The paper also has weak spots: first, the experimental evaluation is very weak. Technically, an ICML reviewer can ignore the appendix per review guidelines. Then only one dataset of very small size is used to evaluate the method. However, even considering the additional results in the appendix, the experiments are too weak. First, this method is evaluated on only 12 datasets and, judging by Appendix B, is the best method in about 50% of the cases. Here a more detailed comparison, possibly with CD-Diagrams is missing. I understand that more experiments / datasets is not always insightful, but when you present a new method, you should show evidence why/where it is better. Additionally, the number of trees is low, which makes pruning uninteresting. Besides the experimental evaluation, the paper also fails to explain the impact of the theoretical analysis. Currently, section 4 reads as "look we also have nice theory"; which is never ever visited again, nor does it provide special guarantees that show impact in the experimental evaluation. Finally, the authors fail to motivate why faithful pruning is important. In the most simplistic case, I can perfectly approximate a tree ensemble with a single tree. While eq. 3 has a slightly different formulation, this tree would be, in the spirit of faithful pruning, a good solution. The authors dismiss this point by stating that "[..] distill the ensemble into a single decision tree, changing the model class; we leave a small-scale comparison to future work.", which I, personally, find very debatable. However, I admit that the authors mainly follow the framework by (Emine etal. 2025) and thus my criticism about why faithful pruning is important should probably be directed towards Emine etal.
In summary, I think the paper has a good core and might be fitting for ICML. However, there are also some weak spots, that might be explained by the rebuttal.

---

> ### Author Rebuttal · Authors · 2026-03-29
>
> Thank you for the detailed and thoughtful review. We answer the three key questions in order.
>
> **1. What is the goal for pruning?**
>
> Our goal is decision-consistent post-hoc compression, not accuracy-oriented pruning: the original ensemble remains the reference model, and the smaller model should keep the original model's predictions unchanged on likely future inputs. This still targets memory/latency gains, but under a backward-compatible requirement. Concretely, this matters for (i) prediction churn/usability, i.e. avoiding unnecessary output changes for similar cases after an update (Milani Fard et al., 2016; Bahri and Jiang, 2021), (ii) human-AI compatibility, i.e. preserving users' learned trust patterns about when the model is reliable (Bansal et al., 2019), (iii) downstream breakage, i.e. not introducing new errors into pipelines that depend on the deployed predictions (Srivastava et al., 2020), (iv) faithful explanation/recourse, i.e. keeping explanations tied to the actual deployed decision rule rather than a surrogate (Rudin, 2019), and (v) verification/global guarantees, i.e. retaining properties one wants to certify for the deployed ensemble over a region of the input space (Devos et al., 2021; Calzavara et al., 2023).
> We agree that M=30/50 is conservative; we chose this as a conservative fully certified benchmark, not because PINE is limited to that scale. Appendix B and our follow-up experiments at n=70/100 and depth-3 n=50 show similar trends, with α=0.8 depth-2 fidelity still about 99.0-99.1% and pruning about 63-67%. See our response to reviewer BebJ, Q1/4, for the fuller larger-ensemble discussion.
>
> **2. What is the appeal behind faithful pruning?**
>
> Because the goal is to keep deployed predictions unchanged, faithful pruning is a different compression target from accuracy-oriented pruning (Emine et al., 2025). We agree that, if one allows a model-class change, a single exact tree is a valid exact alternative; this is precisely the born-again tree (BAT) setting (Vidal and Schiffer, 2020). But "one tree" need not mean "small or cheap": BAT itself reports 567.0 average leaves versus 51.2 in the original forest before post-pruning, and BATree's exact problem is NP-hard with computational time growing exponentially in the number of features. By contrast, Eq. (3) keeps the original ensemble family, the Chow-Liu Oracle adds only O(pB^2) constraints, and as shown in our response to reviewer GhSa, Q2, input dimension had only a weak runtime correlation in our added experiments (Spearman ρ=+0.083). So BAT and faithful pruning are better viewed as complementary tools for different deployment needs. By contrast, pruning in the "Many Could Be Better Than All" sense (Zhou et al., 2002) intentionally allows the decision function to change to improve generalization, which is also important but not backward-compatible compression: as noted in Q1, such prediction changes can themselves be problematic in auditing, explanation/recourse, verification, and other deployment settings where the compressed model is expected to preserve the deployed decision rule.
>
> **3. How does the method perform overall and against broader baselines?**
>
> We completed follow-up experiments and found that PINE remains the strongest method in aggregate fidelity even after expanding the comparison to FIPE + all 15 documented PyPruning methods under the same splits and hyperparameters. In the main 12-dataset x 5-seed XGBoost depth-2 study, PINE-CL increases mean pruning from 44.6% to 67.8% as α goes 0.05->0.8, while mean fidelity changes only from 99.96% to 99.15% and mean accuracy stays near zero. So the aggregate picture is a stronger pruning-fidelity trade-off at very high fidelity.
>
> | method | fidelity | Δ vs PINE |
> |---|---:|---:|
> | PINE (Ours) | 99.57 | 0.00 |
> | cluster_centroids | 96.74 | -2.83 |
> | error_ambiguity | 95.42 | -4.15 |
> | drep | 95.41 | -4.17 |
> | individual_error | 95.23 | -4.34 |
> | largest_mean_distance | 94.31 | -5.27 |
> | reduced_error | 94.22 | -5.36 |
> | individual_kappa_statistic | 92.84 | -6.74 |
> | complementariness | 92.75 | -6.83 |
> | cluster_accuracy | 91.07 | -8.51 |
> | individual_contribution | 90.84 | -8.74 |
> | combined_error | 89.69 | -9.89 |
> | margin_distance | 88.82 | -10.76 |
> | individual_margin_diversity | 88.50 | -11.08 |
> | combined | 86.90 | -12.68 |
> | reference_vector | 86.59 | -12.99 |
>
> Against the best PyPruning method at each matched pruning level, PINE has the highest aggregate mean fidelity at all 6 α settings and wins 63/72 dataset x α comparisons. Relative to FIPE, it improves pruning in 71/72 settings by 21.8 pp on average while keeping fidelity very high. The few losses occur only on Breast-Cancer-Wisconsin and Seeds, the most quantized small datasets in this bundle, where one or two test predictions already move average fidelity visibly. Some PyPruning methods can be competitive on post-pruning accuracy, which is expected because they optimize accuracy/compression rather than fidelity.

---

> > ### Author Rebuttal · Reviewer_JqYT · 2026-04-01
> >
> > All questions were answered adequately, and the authors took the time to run additional experiments.

---

> > > ### Author Response · Authors · 2026-04-06
> > >
> > > We thank the reviewer for taking the time to review our paper and for acknowledging our rebuttal. We are grateful for the reviewer's valuable suggestions, which will help us further strengthen the paper. In the revision, we will specifically:
> > >
> > > 1. **Clarify earlier and more explicitly that PINE targets decision-consistent post-hoc compression**, and **why faithful pruning matters in deployment settings**.
> > > 2. **Make Algorithm 1 more self-contained** by briefly defining the roles of the pruner and the Oracle.
> > > 3. **Strengthen the presentation of the theory** by clarifying how the in-distribution guarantee and Oracle-request bound affect the pruning problem in practice.
> > > 4. **Move the broader empirical evidence into the main paper**, including the expanded comparison against FIPE and PyPruning baselines, clearer aggregate summaries, and the larger-ensemble trends.
> > >
> > > We thank the reviewer again for the thoughtful evaluation.

---

### Official Review · Reviewer_BebJ · 2026-03-11

**Soundness:** 3
**Presentation:** 3
**Significance:** 3
**Originality:** 4
**Overall Recommendation:** 4
**Confidence:** 4

**Summary:**

This paper addresses the challenge of pruning tree ensembles to reduce model size and inference costs while preserving prediction accuracy. Existing faithful pruning methods guarantee that the pruned model’s predictions match the original model’s input space. The proposed PINE (Pruning with In-distribution Guarantees) relaxes the constraint via guaranteeing prediction equivalence only within an "in-distribution" region.

**Compliance With Llm Reviewing Policy:**

Affirmed.

**Final Justification:**

The authors answered some of my concerns, while there are still some remaining unsolved. I would keep the initial socre.

**Key Questions For Authors:**

1. Add a discussion on scalability for larger ensembles.
2. Include a comparison against a simple heuristic baseline.
3. riefly summarize the sensitivity analysis of the plausibility score hyperparameters in the main text.
4. Add a discussion on scalability for larger ensembles.
5. Include a comparison against a simple heuristic baseline.
6. Clarify the status of code availability.
7. No clear interpretation in the better performance is given. Some examples of results or typical data should be given with analysis.

**Limitations:**

While the paper acknowledges that MILP can be computationally expensive, a more systematic discussion on scalability is needed. Although table 1 shows promising runtime reductions for PINE on the Pima-Diabetes dataset as α increases, practitioners need clarity on how the method scales to larger ensembles and higher-dimensional data. The Appendix results for M=50 are a good start, but a direct runtime analysis for larger models would significantly strengthen the practical utility of the work.
Although the appendix compares different scoring mechanisms, the main text relies on the Chow-Liu tree with a specific bin count. A brief discussion in the main body regarding the selection of the plausibility score and the impact of the bin count on the trade-off between coverage, pruning rate, and runtime would provide better practical guidance. Summarizing the key findings from the sensitivity analysis (Appendix B.3) in the main text is recommended.

**Strengths And Weaknesses:**

Strengths
	The paper correctly identifies a limitation in current faithful pruning methods.
	The combination of conformal prediction and MILP-based pruning is sophisticated. The conformal prediction calibrating the region size via a parameter α provides a mechanism for managing the fidelity-compression trade-off. Furthermore, the choice of a Chow-Liu tree for the plausibility score is proper as its additive structure makes it naturally compatible with MILP constraints, avoiding complex non-linear formulations.
	The experimental evaluation is comprehensive and rigorous. Tested on 12 datasets against multiple baselines (FIPE, IC, DREP, MDEP), the results clearly address the research questions. PINE consistently outperforms FIPE in pruning rates, particularly as α increases, while empirically validating the conformal coverage guarantees (see Figure 4). The ablation study comparing different plausibility scores (Chow-Liu, Leaf Support, Isolation Forest) in the appendix further strengthens the validity of the design choices.
	The manuscript is generally well-written and accessible. The problem statement is clear, the preliminaries are sufficient, and the methodology is explained logically. Theoretical results are presented succinctly and provide strong support for the main claims.

Weakness:
	While the shift to in-distribution guarantees is novel, the technical implementation relies heavily on the existing FIPE framework. A more explicit discussion of the specific technical challenges would rise when modifying the FIPE Oracle to incorporate the new in-distribution constraints. Clarifying what goes beyond simply adding MILP constraints would better highlight the technical depth of the contribution.
	The current comparison focuses on sophisticated pruning methods. Including a naive baseline, such as pruning trees based on feature importance or weight magnitude followed by a fidelity check on a validation set, would be valuable.

---

> ### Author Rebuttal · Authors · 2026-03-29
>
> Thank you for the positive and careful review. We answer the key questions in order.
>
> **1 / 4. Add a discussion on scalability for larger ensembles.**
>
> Thank you for raising scalability. Our main benchmark uses a conservative fully certified setting, but our follow-up experiments show that the qualitative pattern extends beyond it. Appendix B.4 includes larger XGBoost ensembles and Random Forests, and we additionally ran experiments with deeper/larger XGBoost models. Across the depth-2 n=70/100 experiments at α=0.8, pruning stays roughly 63-67% while fidelity remains about 99.0-99.1%, and the depth-3 n=50 setting still reaches 47.0% pruning with 99.28% fidelity. At fixed α=0.8, mean fidelity stays around 99% while runtime grows with model size: for (M,D)=(30,2),(50,2),(70,2),(100,2),(30,3),(50,3),(100,3), where (M,D) denotes number of trees and max depth, runtime is 4.29, 165.11, 921.38, 3280.48, 44.03, 1206.33, and 13148.44 s, respectively. So fidelity remains high and pruning substantial as models grow, although reducing absolute runtime for larger ensembles is an important direction for future work. In the revision, we will surface these larger-model results more explicitly in the main text. See our response to reviewer GhSa, Q2, for the fuller runtime discussion.
>
> **2 / 5. Include a comparison against a simple heuristic baseline.**
>
> We completed follow-up experiments against three simple Top-K heuristics and found that PINE remains stronger on fidelity. Each heuristic assigns a simple hand-designed score to each tree and retains the top K trees under that score:
>
> | method | heuristic score | fidelity | Δ vs PINE |
> |---|---|---:|---:|
> | PINE | - | 99.57 | 0.00 |
> | LeafMagnitude-TopK | average absolute leaf value | 95.97 | -3.61 |
> | TreeGain-TopK | total split gain | 95.50 | -4.08 |
> | FeatImpTree-TopK | feature-gain-weighted tree score | 94.41 | -5.16 |
>
> The conclusion is clear: simple heuristics may give a smaller model, but they do not preserve the original ensemble's decisions as well as PINE. The strongest heuristic, LeafMagnitude-TopK, is still 3.61 pp below PINE in average fidelity; on the low-α slice, the heuristics are around 96.5% fidelity with a 0.8-1.2 pp accuracy drop, whereas matched PINE is about 99.95% fidelity with near-zero average accuracy change. This is why we evaluate them against fidelity rather than only test accuracy or model size.
>
> **3. Briefly summarize the sensitivity analysis of the plausible score hyperparameters in the main text.**
>
> Thank you for this suggestion. In the final manuscript, we will summarize in the main text that Appendix B.3 compares Chow-Liu, Leaf Support, and Isolation Forest, and reports coverage, fidelity, pruning rate, and runtime across B=4/8/16 for the Chow-Liu constraint. The practical takeaway is that score choice and bin count mainly trade runtime against coverage and pruning, while the qualitative behavior stays stable across B. We selected Chow-Liu for the main text because it is MILP-friendly and gives the strongest overall fidelity-pruning behavior among the tested scores. In the main setting, empirical coverage tracks 1-α with mean absolute calibration error about 0.64 pp, and conditional fidelity within the in-distribution region remains 100%. We will present this as practical guidance on the runtime/coverage/pruning tradeoff, so readers can see how score and bin choices affect use in practice.
>
> **6. Clarify the status of code availability.**
>
> If accepted, we must release the code.
>
> **7. No clear interpretation of the better performance is given.**
>
> Thank you for pointing this out. We will revise this discussion to make the mechanism more concrete. In Fig. 1, the faithful baseline must preserve decisions over the whole input space, corresponding to the full blue region in panel (b), whereas PINE only needs to preserve decisions over the calibrated in-distribution region, the blue region in panel (c). As a result, PINE only needs to describe the black decision boundary within the region aligned with the data distribution, which can be done with fewer trees than representing the boundary everywhere, including low-density boundary fragments. We will add more explicit visual examples of this mechanism for both a synthetic example and a real-data case study in the appendix/supplement to make the explanation easier to inspect.
>
> In our focused follow-up case study on the Adult and COMPAS datasets, the same pattern appears quantitatively: PINE concentrates counterexample search in regions that are closer to the observed data manifold and higher-margin than those emphasized by the faithful baseline, with mean kNN-Gower distance 0.084 vs 0.321 and mean margin 0.480 vs 0.345. Intuitively, PINE spends its equivalence budget on future inputs from the same distribution rather than on far-OOD boundary fragments. We will revise the text to make clearer that the added technical step is calibrated in-distribution prediction equivalence within the FIPE framework.

---

> > ### Author Rebuttal · Reviewer_BebJ · 2026-04-01
> >
> > In general, the authors responseed with providing more experiments. But the focus on interpretability is not settled. In my opinion, the general evaluation is not changed because I can't see the promise to improve the overall quality of the paper.

---

> > > ### Author Response · Authors · 2026-04-06
> > >
> > > Thank you for raising the interpretability issue. We took this point as highlighting that the manuscript did not clearly explain why PINE achieves a better pruning-fidelity trade-off than FIPE, and in this final response we clarify that mechanism through further analysis and two concrete case studies.
> > >
> > > **Qualitative explanation: Why PINE outperforms FIPE.**
> > >
> > > PINE achieves a better pruning-fidelity trade-off than FIPE because, once the guarantee is restricted to likely future inputs from the same distribution, some trees that only protect OOD regions can be efficiently pruned. In Fig. 1, FIPE must preserve decisions over the whole input space, so the Oracle keeps searching for counterexamples even in low-density OOD regions. PINE changes that search space: the plausible-score threshold, calibrated by split conformal prediction, restricts Oracle search space to a calibrated in-distribution region. As a result, PINE only needs to preserve the decision boundary aligned with the data distribution rather than boundary regions everywhere. Under FIPE, a tree may be retained only to avoid violating prediction equivalence on an OOD point, even if it does not matter on typical data.
> > >
> > > **Quantitative evidence.**
> > >
> > > Most baseline FIPE counterexamples exist outside PINE's calibrated region, and as α increases, PINE needs to preserve far fewer counterexamples. These are the main quantitative findings supporting the qualitative mechanism above. The pooled Adult/COMPAS comparison below summarizes these sets by kNN-Gower distance and prediction margin.
> > >
> > > |Metric|PINE|FIPE|Insight|
> > > |---|---|---|---|
> > > |mean kNN-Gower|0.084|0.321|PINE focuses on more in-distribution counterexamples, whereas FIPE focuses on more OOD counterexamples.|
> > > |mean margin|0.480|0.345|PINE focuses on regions where the original ensemble prediction is more stable, whereas FIPE focuses on less stable OOD regions.|
> > >
> > > We then analyzed the Adult dataset directly by rescoring FIPE counterexamples under PINE's calibrated region and tracking how many PINE counterexamples remain as α increases. This Adult-dataset analysis makes the constraint-filtering mechanism explicit. First, in the conservative setting α=0.05, all 535/535 FIPE counterexamples (oracle counterexamples found by baseline FIPE) fall outside PINE's calibrated region. Second, as α increases to 0.6, the number of remaining PINE counterexamples (oracle counterexamples found under PINE's calibrated in-distribution guarantee) shrinks from 442 to 18, while mean kNN-Gower drops from 0.097 to 0.051. This supports the same interpretation: PINE achieves a better pruning-fidelity trade-off than FIPE because it removes prediction-equivalence constraints on atypical points and concentrates pruning on more in-distribution regions.
> > >
> > > ---
> > >
> > > Additionally, we conducted case studies on concrete points that baseline FIPE still treats as oracle constraints but PINE excludes as OOD points. These examples make the excluded OOD structure concrete. Excluding such points allows PINE to solve the pruning problem with fewer added constraints and likely prune more trees.
> > >
> > > **Case study: Adult.**
> > >
> > > The Adult case shows a point with clearly implausible feature combinations, including two inconsistent pairs: `{education level=Preschool, years of education=13.50025}`; `{marital status=Never-married, relationship role=Wife}`. These pairs describe the same underlying status in incompatible ways, so this is not a plausible rare person from the data distribution. Its kNN-Gower distance is 0.383 and its margin is 0.514, so it is a confident OOD counterexample rather than a point close to the calibrated in-distribution boundary. FIPE must still preserve prediction equivalence on such points, whereas PINE can exclude them.
> > >
> > > **Case study: COMPAS.**
> > >
> > > The COMPAS case shows a point with logically inconsistent category assignments. It simultaneously indicates `{number of prior offenses=0, number of prior offenses=>3}`, even though these categories are mutually exclusive by construction, and it also has `{juvenile crimes=0, juvenile felonies!=0, juvenile misdemeanors!=0}`, which is inconsistent with the zero-count juvenile indicators. So this is not simply unusual; it violates the internal logic of the binarized representation itself. Its kNN-Gower distance is 0.500. FIPE must still preserve prediction equivalence on such points, whereas PINE can exclude them; at the run level, this is accompanied by a drop from 12.0 to 4.6 mean oracle iterations and from 18.8 to 10.8 mean retained trees on COMPAS.
> > >
> > > We will make this mechanism explicit in the paper by adding a main-text paragraph around Fig. 1 and a concise Adult/COMPAS appendix case study with representative in-distribution-region and OOD-region examples, as well as the corresponding changes in oracle iterations and retained trees. We sincerely appreciate the reviewer's valuable comment.

---

### Official Review · Reviewer_PfxG · 2026-03-13

**Soundness:** 4
**Presentation:** 4
**Significance:** 3
**Originality:** 3
**Overall Recommendation:** 5
**Confidence:** 4

**Summary:**

PINE: Pruning Boosted Tree Ensembles with
Conformal In-Distribution Prediction Equivalence
This paper studies the tradeoff between fidelity and compression in boosted tree ensembles when the pruning algorithm only has to match the original within a known in-distribution region. They propose an approach to adapting conformal prediction to a previous pruning approach. This is done by training Chow Liu Trees to output a likelihood based scoring function for examples which is thresholded on a validation set. They show experimental results indicating this approach can prune ensembles better than several baselines while maintaining high fidelity.

**Compliance With Llm Reviewing Policy:**

Affirmed.

**Final Justification:**

The authors have adequately responded to all my concerns.

**Key Questions For Authors:**

see weaknesses above

**Strengths And Weaknesses:**

Strengths:
+ Tree ensembles are often an approach of choice is many problems, so pruning them is an interesting problem.
+ The paper is very well written and the ideas are clearly explained.
+ This is a nice use of Chow Liu trees. In general the idea is very simple, but elegant.
+ Good results on a wide range of problems.

Weaknesses:
- It is not clear to me how the alpha parameter could be chosen in practice. It is not clear how this correlates with actually being "in distribution". This is a key weakness, as without a meaningful way to choose alpha it is not clear how this approach could be useful.
- FIPE, on which this approach is heavily based, is not described in enough detail. The paper is not self contained (without the appendix).
- In experiments, a max depth of 2 and ensemble size 30 is used. These are already very small ensembles.
- It is confusing why the FIPE results are shown between 15-20% pruning rates.
- Would there be any pruning relative to FIPE if the MILP was solved setting alpha to zero? In other words, what is the contribution to the pruning just of the Chow Liu Tree based scoring (or some scoring)?

To summarize, this is a good paper exploring an interesting topic. It is well written and the results are clear. However, it is not clear how this could be used in practice.

---

> ### Author Rebuttal · Authors · 2026-03-29
>
> Thank you for the clear review. We answer the five questions in order.
>
> **1. How should α be chosen in practice?**
>
> In PINE, α is the user-set split-conformal miscoverage level. Practically, α=0.05 or 0.1 means asking that predictions remain unchanged on about 95% or 90% of future inputs from the same distribution. Under exchangeability, α therefore specifies the tolerated probability that a future input falls outside the calibrated in-distribution region. The "in-distribution" part comes from the plausible score and calibrated threshold; α itself is not a per-point OOD score. Empirically, the calibrated region tracks the target closely: 95.38, 90.53, 80.67, 60.50, 40.41, and 21.34% coverage at α=0.05, 0.1, 0.2, 0.4, 0.6, and 0.8, respectively, with mean calibration error about 0.64 pp and 100% conditional fidelity inside this region. Operationally, smaller α gives a broader, more conservative calibrated region; larger α gives a smaller region and allows stronger compression.
> A practical default is to start with α=0.05 or 0.1 when high decision consistency is required, and increase α only if more compression is worth a narrower region. So α is less a heuristic tuning knob than an explicit coverage target.
> Users can also check whether a specific input lies in this region, so practically important cases can be inspected one by one. Under distribution shift, this guarantee can weaken, so monitoring or recalibration may be needed. We will clarify this interpretation in the paper and mention utility-aware α selection only as future work at present.
>
> **2. FIPE is not described enough / not self-contained.**
>
> Thank you for flagging this. In the final manuscript, we will make Section 3.2 self-contained by briefly restating the FIPE Pruner-Oracle loop around Algorithm 1 and defining the Pruner and Oracle there. FIPE alternates the Pruner and Oracle on a finite constraint set: the Pruner updates the weights on the current set, and the Oracle either certifies prediction equivalence or returns a counterexample to add. PINE keeps this pruning step unchanged and only restricts the Oracle search to the calibrated in-distribution region. This will also make the technical delta clearer: Chow-Liu is the score we use because it is both statistically meaningful and MILP-compatible.
>
> **3. Are the ensembles too small?**
>
> The main text starts from a conservative setting, but the qualitative benefit is not confined to it. Appendix B.4 shows similar trends for larger XGBoost ensembles (M=50) and Random Forests, and our follow-up experiments extend this to depth-2 n=70/100 and depth-3 n=50. At α=0.8, pruning stays roughly 63-67% at n=70/100 while fidelity remains about 99.0-99.1%, and the depth-3 n=50 setting still reaches 47.0% pruning with 99.28% fidelity. So the default benchmark is modest, but the same qualitative conclusion extends to materially larger settings. We will add this evidence to the final manuscript; see our response to reviewer BebJ, Q1/4, for the fuller larger-ensemble discussion and our response to reviewer GhSa, Q2, for the fuller runtime discussion.
>
> **4. Why does FIPE appear around 15-20% pruning?**
>
> This was our presentation issue, and we will fix the figure caption and surrounding text. It is not a second FIPE sweep. FIPE is not a pruning-rate-controlled curve; for each trained ensemble and seed it returns a single sparsest faithful solution, whereas PINE produces multiple operating points by sweeping α and IC/DREP/MDEP produce curves because the target number of retained trees is given as input. In the main-paper figure, the 15-20% number is the Pima-Diabetes example: Table 1 reports FIPE pruning of 17.3% there. By contrast, in the full 12-dataset aggregate the faithful baseline averages 35.2% pruning. We will state explicitly that FIPE contributes a single-point baseline there.
>
> **5. What happens as α approaches zero?**
>
> At α=0, there should be no extra pruning gain from the Chow-Liu score alone. As α approaches zero, the calibrated region expands back toward the faithful case; at the exact endpoint, split conformal gives τ(0)=+∞, so X_ID(0)=X and PINE reduces to FIPE. In that limit the score no longer shrinks the Oracle search region, so any residual difference would be implementation-level rather than a separate scoring contribution. In other words, the gain comes from restricting the region, not from the Chow-Liu score by itself. This is also why the gap is already much smaller at small positive α: in the 12-dataset x 5-seed main setting, the faithful baseline averages 35.2% pruning, while PINE at α=0.05 averages 44.6% pruning with 99.96% fidelity. The gain is already much smaller near the faithful case than at large α, exactly as expected if region restriction is the source and it disappears at α=0.

---

> > ### Author Rebuttal · Reviewer_PfxG · 2026-04-03
> >
> > I thank the authors for their response. Most of my concerns have been addressed. I am still concerned about the choice of alpha. I understand that the algorithm will more or less calibrate the in-distribution region correctly. But given that alpha=0 results in no pruning, the suggested alpha=0.05 seems minimal. I was looking for something more principled, e.g. if the scores are properly calibrated, could they be used somehow to choose the largest alpha for say 95% fidelity? A fidelity target seems much more practical than a coverage target.

---

> > > ### Author Response · Authors · 2026-04-06
> > >
> > > We thank the reviewer very much for the thoughtful follow-up comments. In this response, we clarify how practical α selection can be done post hoc by conducting additional experiments that directly evaluate which α values are selected under fidelity-targeted criteria.
> > >
> > > **Practical α-selection protocol.**
> > >
> > > We conducted additional experiments on practical post-hoc α selection by introducing a held-out selection split for choosing α after pruning. Specifically, we changed the usual three-way data split to `fit/calibration/selection/test` and compared α candidates using fidelity to the original ensemble on the held-out selection split. To study how α should be chosen in practice, we implemented and compared the following two selection rules:
> > >
> > > |Selector|Rule|
> > > |---|---|
> > > |Empirical selector|Among the α candidates, choose the largest one whose fidelity on the held-out selection subset meets the target fidelity, where target fidelity denotes the desired fidelity level.|
> > > |Confidence-bound-based selector|Among the α candidates, choose the largest one whose held-out mismatch risk is certified to be below `1 - target fidelity`. Concretely, we use a one-sided Clopper-Pearson upper bound with Bonferroni correction over the 8 α candidates; if no positive α qualifies, we fall back to the unpruned ensemble.|
> > >
> > > The former empirical selector follows the same basic idea as LOP (Devos et al., 2025), in that both use a held-out split and choose the largest compression level that satisfies an empirical fidelity/performance constraint. The latter confidence-bound-based selector follows the idea of Learn-then-Test (Angelopoulos et al., 2025), in that it adds a finite-sample statistical acceptance rule over a finite candidate grid.
> > >
> > > **Experimental setting.**
> > >
> > > We conducted these additional experiments on the 12 datasets with (M,D)=(30,2), where M is the number of trees and D is the maximum depth. We used a fit/calibration/selection/test split of 48/16/16/20 and the candidate grid α ∈ {0.05, 0.1, 0.2, 0.4, 0.6, 0.8, 0.9, 0.95}. We evaluated both selectors at target fidelity 95% and 99%. The selection split data was used only for post-hoc selection among α-specific pruned models.
> > >
> > > **Results.**
> > >
> > > The additional experiments suggest that practical α should be selected post hoc by a held-out fidelity rule rather than fixed in advance as a user-set value, because the selected α varies across datasets and target fidelity levels. We summarize the results in the table below, where "FB" denotes reverting to the unpruned ensemble.
> > >
> > > |Target fidelity [%]|Selector|Selected α / FB across 12 datasets|Mean pruning rate [%]|Mean test fidelity [%]|Insight|
> > > |---|---|---|---|---|---|
> > > |95|Empirical|α=0.95 on 12/12 datasets|70.8|98.77|α=0.95 itself satisfies the held-out 95% criterion on all 12 datasets.|
> > > |95|Confidence-bound-based|α=0.95 on 8/12 datasets, α=0.4 on 1/12, α=0.2 on 2/12, FB on 1/12|57.2|99.72|Even the conservative rule keeps α high on most datasets while maintaining strong pruning.|
> > > |99|Empirical|α=0.95 on 8/12 datasets, α=0.6 on 1/12, α=0.4 on 1/12, α=0.2 on 2/12|61.1|98.76|Even at the stricter target fidelity level, the empirical rule still selects moderate-to-large α values on most datasets.|
> > > |99|Confidence-bound-based|α=0.95 on 7/12 datasets, α=0.4 on 1/12, FB on 4/12|39.2|99.96|At the stricter 99% target, FB becomes more frequent and the mean pruning rate decreases to 39.2%.|
> > >
> > > **Summary.**
> > >
> > > The additional experiments show that, for practical use, α is better selected post hoc using held-out data than fixed in advance as a user-set value. To clarify how α should be interpreted in PINE in light of these experiments, we emphasize that α has two roles:
> > >
> > > 1. α acts as the split-conformal miscoverage level that determines the size of the certified in-distribution region X_ID(α).
> > > 2. α determines which pruning level is used in practice.
> > >
> > > The first role answers the coverage question, but it does not by itself tell us which α should be chosen to maximize compression while keeping global prediction equivalence with the original ensemble above a desired level. Calibrated plausibility scores alone do not determine that practical choice, because each α induces a different pruning problem and produces a newly optimized pruned model.
> > >
> > > Under this interpretation, our additional α-selection protocol directly addresses the reviewer's concern. The empirical selector is the direct held-out selection rule, and the confidence-bound-based selector is a conservative finite-grid variant in the same spirit. Note that the test-set fidelities reported above should be read as empirical downstream outcomes of the held-out selection rule, not as formal guarantees.
> > >
> > > In the revision, we will add the practical α-selection discussion summarized above so the paper more clearly separates the conformal role of α from post-hoc fidelity-based α selection. We are very grateful to the reviewer for this valuable suggestion for practical α selection.

---

### Decision · Program_Chairs · 2026-04-30

**Decision:**

Accept (regular)

**Comment:**

This paper introduces PINE, a novel pruning method for tree ensembles that achieves superior compression ratios while maintaining prediction equivalence within a conformal-calibrated in-distribution region. By leveraging conformal calibration to control the equivalence region, PINE offers a more favorable trade-off between compression and prediction consistency compared to existing faithful pruning techniques. The submission received generally positive reviews, with three reviewers supporting acceptance and the fourth (GhSa) expressing initial concerns that were effectively addressed in the rebuttal through additional experiments. Given its technical merit, clear presentation, and strong empirical results, I recommend an acceptance for this submission. In addition, it is recommended that the authors incorporate the additional experimental results and discussions on practical $alpha$ selection into the camera-ready version to further enhance the paper's impact.